# Ribonuclease 4 functions as an intestinal antimicrobial protein to maintain gut microbiota and metabolite homeostasis

Jun Sun[1,2,3,9], Muxiong Chen[1,9], Zhen Hu[1,2,3,9], Ningqin Xu[4,9], Wenguang Wang[1,2,3], Zejun Ping[1,2,3], Jiayi Zhu[1], Desen Sun[5], Zhehao Zhu[6], Hangyu Li[6], Xiaolong Ge[1], Liang Luo[7], Wei Zhou[1], Rongpan Bai [1,10] ✉, Zhengping Xu [1,2,3,10] ✉ & Jinghao Sheng [1,2,3,8,10] ✉

Antimicrobial proteins contribute to host-microbiota interactions and are associated with inflammatory bowel disease (IBD), but our understanding on antimicrobial protein diversity and functions remains incomplete. Ribonuclease 4 (Rnase4) is a potential antimicrobial protein with no known function in the intestines. Here we find that RNASE4 is expressed in intestinal epithelial cells (IEC) including Paneth and goblet cells, and is detectable in human and mouse stool. Results from *Rnase4*-deficient mice and recombinant protein suggest that Rnase4 kills *Parasutterella* to modulate intestinal microbiome, thereby enhancing indoleamine-2,3-dioxygenase 1 (IDO1) expression and subsequently kynurenic and xanthurenic acid production in IECs to reduce colitis susceptibility. Furthermore, deceased RNASE4 levels are observed in the intestinal tissues and stool from patients with IBD, correlating with increased stool *Parasutterella*. Our results thus implicate Rnase4 as an intestinal antimicrobial protein regulating gut microbiota and metabolite homeostasis, and as a potential diagnostic biomarker and therapeutic target for IBD.

The mammalian intestine harbors a diverse and dynamic microbial community crucial to various physiological functions, including digestion and immune system development[1,2]. Perturbation of the microbiota composition (i.e., dysbiosis) is linked to chronic gastrointestinal diseases, such as inflammatory bowel disease (IBD)[3]. Several bacterial taxa are associated with IBD, including *Parasutterella*, an obligate anaerobic, non-spore-forming, gram-negative bacterium recognized as a member of the core gut microbiota[4–6]. Interestingly, *Parasutterella* colonization in mice significantly alters the intestinal metabolome without drastically shifting the overall microbial structure, particularly affecting the kynurenine pathway of tryptophan metabolism[4]. However, *Parasutterella* itself lacks key tryptophan metabolism-related genes in its genome, suggesting that its regulation of host tryptophan metabolism occurs through a paracrine manner. The indoleamine-2,3-dioxygenase 1 (IDO1), expressed in intestinal epithelial cells and immune cells, plays a critical role in intestinal health by catalyzing the conversion of tryptophan to kynurenine and other downstream metabolites[7]. These metabolites have

[1]Institute of Environmental Medicine and Department of General Surgery, Sir Run Run Shaw Hospital, Zhejiang University School of Medicine, Hangzhou 310058, China. [2]Liangzhu Laboratory, Zhejiang University, Hangzhou 310012, China. [3]Cancer Center, Zhejiang University, Hangzhou 310058, China. [4]Division of Health Sciences, Hangzhou Normal University, Hangzhou 310015, China. [5]Department of Biochemistry and Molecular Biology, and Zhejiang Key Laboratory of Pathophysiology, Medical School of Ningbo University, Ningbo, Zhejiang 315211, China. [6]College of Life Science, Zhejiang University, Hangzhou 310058, China. [7]Department of Gastroenterology, Sir Run Run Shaw Hospital, Zhejiang University School of Medicine, Hangzhou 310058, China. [8]Affiliated Hangzhou First People's Hospital, Zhejiang University School of Medicine, Hangzhou 310006, China. [9]These authors contributed equally: Jun Sun, Muxiong Chen, Zhen Hu, Ningqin Xu. [10]These authors jointly supervised this work: Rongpan Bai, Zhengping Xu, Jinghao Sheng. ✉e-mail: rpbai@zju.edu.cn; zpxu@zju.edu.cn; jhsheng@zju.edu.cn

immunomodulatory effects and contribute to maintaining intestinal homeostasis, with imbalances in their production being linked to IBD[8,9]. Despite these findings, the mechanisms regulating *Parasutterella* and its involvement in the pathogenesis of IBD remain unclear.

The intestinal epithelium produces and secretes a broad range of antimicrobial proteins, such as defensins[10–12], cathelicidins[13,14], and Reg3 family[15,16], into the lumen to preserve microbiota balance[17]. These proteins have distinct mechanisms of action and targets, allowing them to effectively combat the challenges posed by complex bacterial populations[18,19]. Despite the identification of several classes of antimicrobial proteins, the full repertoire of these factors in the gut remains to be fully characterized. The vertebrate-specific ribonuclease A (RNASEA) superfamily comprises a series of proteins with antimicrobial activities that play crucial roles in host defense[20]. Their high isoelectric point (pI) and amphiphilicity enable these proteins to interact with bacterial membranes, leading to membrane agglutination, permeabilization, and leakage, ultimately resulting in bacterial death[21,22]. These superfamily members have a tissue-specific distribution and exhibit distinct bactericidal activities; for example, RNASE2 and RNASE3 are mainly expressed in eosinophils with antivirus and antigram-negative bacterium roles, respectively[23,24]. RNASE7 is expressed in various epithelial tissues, such as skin and the genitourinary tract, exhibiting broad-spectrum antimicrobial activity[25]. RNASE9 is specifically expressed in the epididymis, with antibacterial activity[26]. According to our online database analysis, members of RNASE1, 4, 5, and 6 are highly expressed in the intestine; however, only RNASE4 and 5 are present in intestinal epithelial cells (Supplementary Fig. 1a, b), suggesting that these two proteins may play more specialized roles in defending against gut bacteria. We previously demonstrated that RNASE5 (also known as angiogenin/ANG) is secreted into the gut lumen and balances the populations of α-Proteobacteria and Lachnospiraceae through its antimicrobial activity[27]. As a highly cationic protein (pI = 9.18), RNASE4 has the potential to bind to negatively charged bacterial membranes; therefore, we hypothesize that RNASE4 may directly act on gut microbes to regulate the microbiota. However, the role of RNASE4 in the intestine remains to be explored.

In this study, we investigate the function of Rnase4 in the gut and its potential involvement in the pathogenesis of IBD. We demonstrate that Rnase4 acts as an antimicrobial protein, modulating the gut microbiota composition by specifically targeting and killing *Parasutterella*, thereby enhancing the production of kynurenic and xanthurenic acid through the IDO1 pathway and ultimately reducing colitis susceptibility. Furthermore, we observe decreased RNASE4 levels in the intestinal tissues and stool samples from patients with IBD, correlating with an increased *Parasutterella* abundance. These findings provide novel insights into the complex interplay between antimicrobial proteins, the microbiota, and intestinal health, highlighting the potential of RNASE4 as a diagnostic biomarker and therapeutic target for IBD.

## Results

### *Rnase4* is expressed in intestinal epithelium and detected in stool samples

Our online database analysis suggested that RNASE4 was mainly expressed in the intestine (Supplementary Fig. 1a, b); therefore, we commenced our study by examining its expression profile in intestinal tissues and stool samples. The online single-cell RNA sequencing data analysis indicated that both mouse and human *Rnase4* transcripts were highly expressed in epithelial cell lineages, including goblet and Paneth cells (Fig. 1a, b). To validate these findings, we isolated different cell types from the mouse intestine and confirmed that *Rnase4* was indeed predominantly expressed in goblet and Paneth cells compared with other epithelial cells (Fig. 1c). We then conducted immunofluorescence staining in mouse tissue and organoids, and found that

the Rnase4 protein was present in the intestinal epithelium, with a notable enrichment in secretory, granule-like structures of goblet and Paneth cells (Fig. 1d, e). Furthermore, we detected the presence of Rnase4 protein in stool samples and the colonic mucus layer via immunoblotting (Fig. 1f). Similar observations were made in human intestinal tissues and stool samples (Fig. 1g, h). Together, these results indicate that Rnase4 is expressed in intestinal epithelial cells, with a predominant localization in secretory epithelial cell lineages, and can be detected in stool samples, implying its potential interactions with intestinal microbiota.

### Mice lacking *Rnase4* have an altered intestinal microbiota

To investigate the potential role of Rnase4 in the intestine, *Rnase4* (*Rnase4*[−/−]) knockout mice were generated using a TALEN-based knockout strategy (Supplementary Fig. 2a), which was verified through genome sequencing and protein expression analysis (Fig. 1d, f, and Supplementary Fig. 2b, c). *Rnase4*[−/−] mice were born in a normal Mendelian ratio, were healthy when reared in a specific pathogen-free facility, and showed normal intestinal morphology with no noticeable changes in the crypt-villus architecture (Supplementary Fig. 3a–c). Moreover, there were no significant differences in the intestinal cell composition—including transit amplifying, Paneth, and goblet cells—, mucus layer thickness, the morphology and cell polarity of Paneth and goblet cells, and the levels of known gut antimicrobial peptides or proteins between wild-type (WT) and *Rnase4*[−/−] mice (Supplementary Fig. 3e–g and Supplementary Fig. 4a–e).

We then analyzed the gut microbiota in both mice. Although overall stool bacterial loads in WT and *Rnase4*[−/−] littermates were similar (Supplementary Fig. 5a), 16 S rDNA sequencing analysis revealed that *Rnase4*[−/−] mice had a less diverse and distinct stool bacterial community compared with WT mice (Fig. 2a, b). Dissimilarity analysis also showed that the microbiome difference between the two genotypes was significantly greater than the variation within each genotype (Fig. 2c). Taxonomic composition and relative abundance analysis revealed that the overall gut microbiome was dominated by the phyla Bacteroidetes and Firmicutes and their subordinate classifications in both groups. However, due to the high species diversity within these two phyla and the variability within each group, no statistically significant differences were observed. Significant differences were identified in the lower abundance phyla and their subordinate classifications, specifically Deferribacteres (genus *Mucispirillum*, upregulated in *Rnase4*[−/−] mice), Proteobacteria (genus *Parasutterella*, upregulated in *Rnase4*[−/−] mice), and Verrucomicrobia (genus *Akkermansia*, downregulated in *Rnase4*[−/−] mice) (Supplementary Fig. 5b–f). To further identify the bacterial taxa with differential abundance, we employed linear discriminant analysis effect size (LEfSe), which comprehensively considers statistical significance, biological consistency, and effect size. This analysis identified several differences in the baseline stool microbiota composition, with the *Mucispirillum* and *Parasutterella* genera being most enriched in *Rnase4*[−/−] mice (Fig. 2d, and Supplementary Table 1). These results were confirmed by quantitative PCR analysis (Fig. 2e) and fluorescence in situ hybridization (Fig. 2f, g).

### *Rnase4* deficiency exacerbates mouse colitis

The strong link between dysbiosis and intestinal inflammation prompted us to investigate the effect of Rnase4 on mouse colitis. Thus, a murine colitis model was established by treating the mice with dextran sulfate sodium (DSS). The results showed that *Rnase4*[−/−] mice exhibited increased susceptibility to DSS induction compared to their WT counterparts, as evidenced by greater weight loss, increased disease activity index, shorter colon length, enhanced intestinal permeability, and more severe histopathological and cytokine level changes in the colons of *Rnase4*[−/−] mice (Fig. 3a–h). To confirm our observations, we examined the effect of Rnase4 in another colitis model—i.e.,

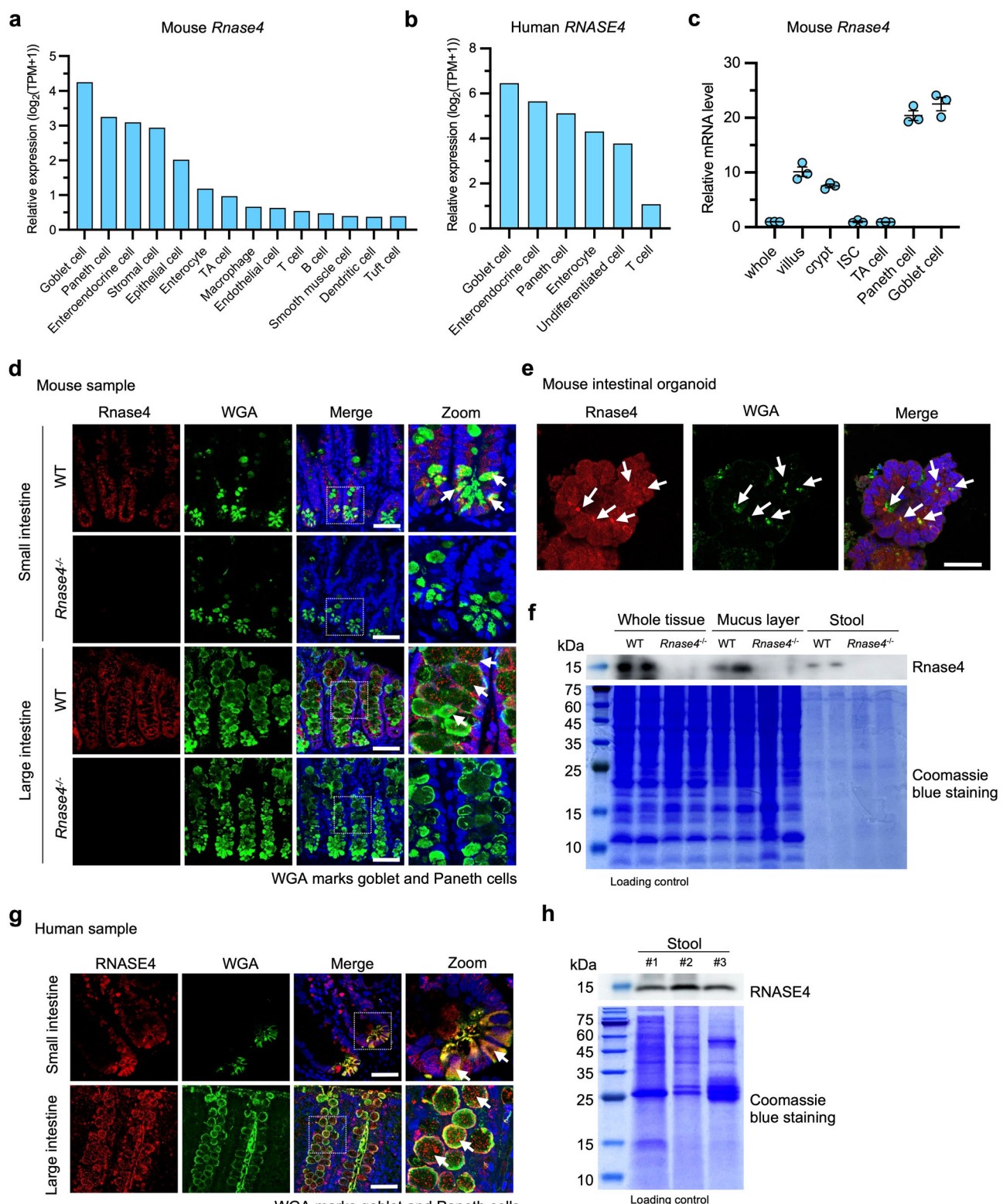

**Fig. 1 | *Rnase4* is expressed in intestinal epithelium and detected in stool samples. a, b** *Rnase4* expression levels in different intestinal cell types based on single-cell RNA sequencing data, sourced from Mouse Cell Atlas (bis.zju.edu.cn/MCA) and Human Protein Atlas (www.proteinatlas.org). **c** *Rnase4* expression levels in different mouse intestinal epithelial cell types by quantitative PCR ($n = 3$ mice). ISC intestinal stem cell, TA transit-amplifying. **d, e** Rnase4 localization in small and large intestines of WT and *Rnase4^{-/-}* mice, and intestinal organoids cultured from WT by immunofluorescence staining. The nuclei were stained with Hoechst 33342. The white arrows indicate the localization of Rnase4 in secretory, granule-like structure of goblet and Paneth cells. Scale bar, 25 µm. **f** Rnase4 protein levels in colon, mucus layer, and stool sample of WT and *Rnase4^{-/-}* mice by immunoblotting. Coomassie blue staining shows the total protein loading amount. **g** RNASE4 localization in human small and large intestines by immunofluorescence staining. The nuclei were stained with Hoechst 33342. The white arrows indicate the localization of RNASE4 in secretory, granule-like structure of goblet and Paneth cells. Scale bar, 25 µm. **h** RNASE4 protein levels in three human stool samples by immunoblotting. Coomassie blue staining shows the total protein loading amount. Data are presented as mean ± SEM (**c**). Representative images from three independent experiments are shown (**d**, **e** and **g**).

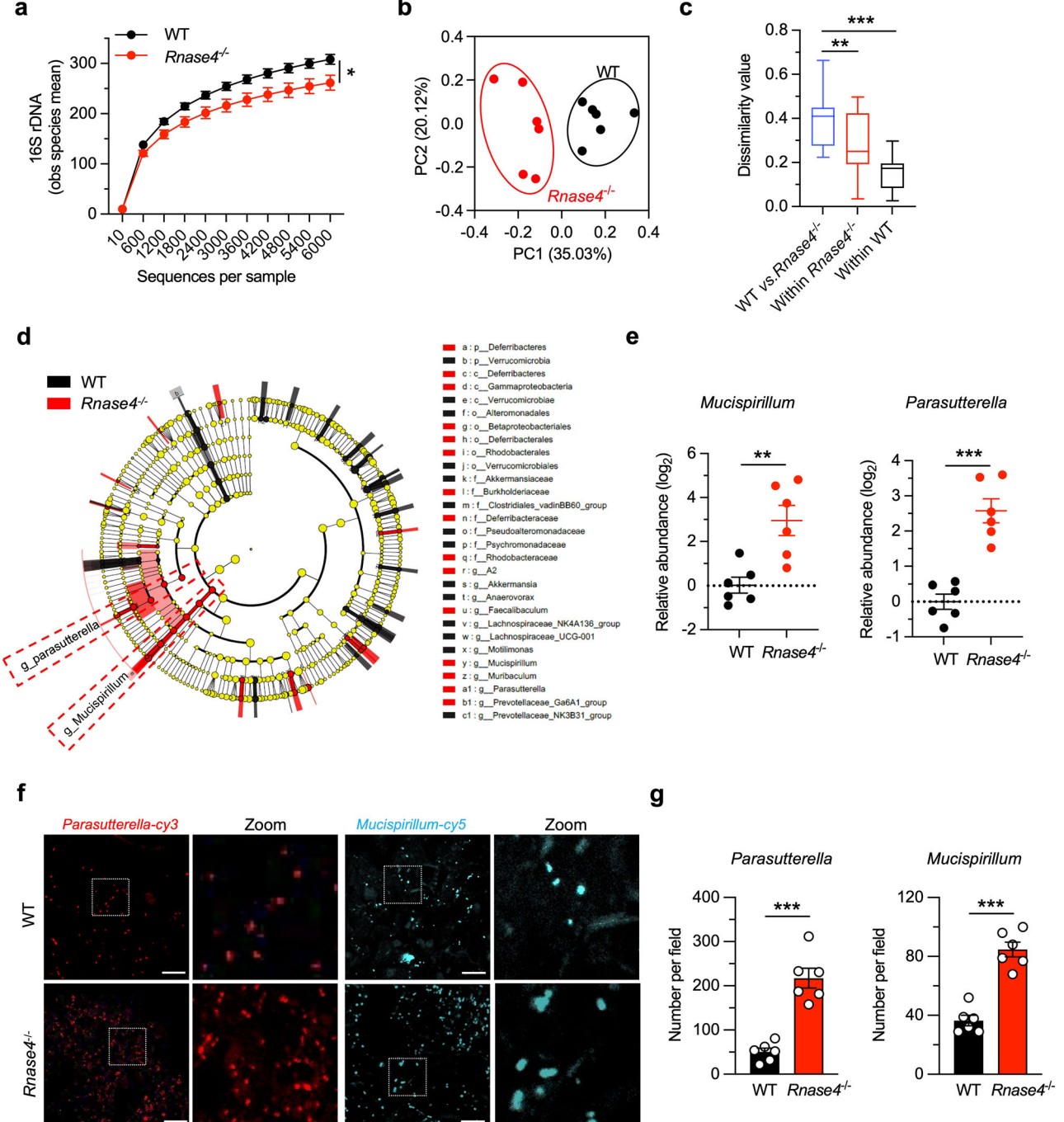

**Fig. 2 | Mice lacking *Rnase4* have an altered intestinal microbiota. a** High-throughput 16 S rDNA sequencing of stool bacterial DNAs from WT and *Rnase4⁻/⁻* mice. The *y*-axis represents the mean of the observed species, indicative of bacterial diversity within each group (*n* = 6). **b** Unweighted UniFrac principal-coordinate analysis of the β-diversity of microbiota composition in WT or *Rnase4⁻/⁻* mice (*n* = 6). **c** Quantification of unweighted UniFrac distance in (**b**). The boxplot represents the minimum value, first, median, and third quartiles, and maximum value. **d** Linear discriminant analysis effect size showing the most differentially abundant taxa of the gut microbiota between WT and *Rnase4⁻/⁻* mice (*n* = 6). **e** The abundance of *Mucispirillum* and *Parasutterella* in the gut microbiota of WT or *Rnase4⁻/⁻* mice by quantitative PCR analysis (*n* = 6). **f** Fluorescent in situ hybridization of colonic lumen sections from WT or *Rnase4⁻/⁻* mice showing the presence of *Parasutterella* (red) and *Mucispirillum* (cyan) in the lumen. Scale bar, 25 µm. **g** Analysis of the number of *Parasutterella* and *Mucispirillum* in the lumen. At least 10 sections per mouse were analyzed (*n* = 6). Data are presented as mean ± SEM; * $p < 0.05$; ** $p < 0.01$; *** $p < 0.001$ by two-tailed unpaired Student's *t*-test (**a**, **e**, and **g**) or two-way analysis of similarities (ANOSIM) test (**c**).

an acute 2,4,6-trinitrobenzene sulfonic acid solution (TNBS)-induced colitis model—and obtained similar results (Supplementary Fig. 6a–g).

Given that Rnase4 is predominantly expressed in intestinal epithelial cells, we employed CRISPR–Cas9 mediated gene targeting to create *Rnase4* conditional knockout (*Rnase4ᶠˡ/ᶠˡ*) mice. Subsequently, these mice were crossed with *Villin-cre* mice to generate intestinal epithelial-specific knockout (*Rnase4ᐃᴵᴱᶜ*) mice (Supplementary Fig. 7a, b). Quantitative PCR and immunoblotting analyses confirmed that Rnase4 levels were significantly reduced in the intestinal epithelium and stool samples of *Rnase4ᐃᴵᴱᶜ* mice, suggesting that the majority of Rnase4 in the stool is secreted from the intestinal epithelium (Supplementary Fig. 7c, d). Consistent with the findings in whole-body

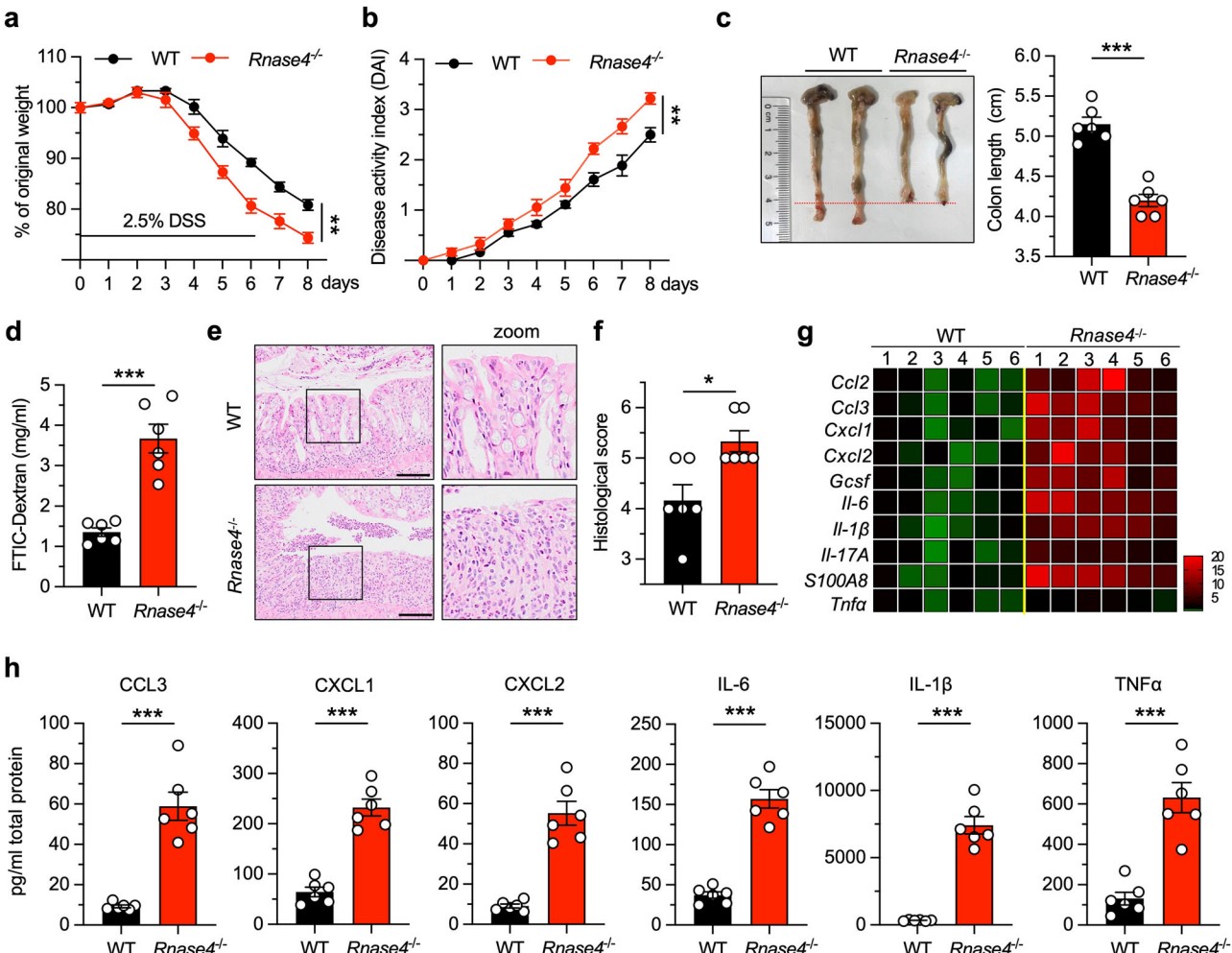

**Fig. 3 | *Rnase4* deficiency exacerbates mouse colitis. a, b** Body weight loss (**a**) and disease activity index (**b**) of WT and *Rnase4⁻/⁻* mice during 2.5% DSS treatment (*n* = 6). **c–f** Colon length (**c**), serum fluorescein isothiocyanate-dextran level (**d**), representative H&E staining image (**e**) and histological score of colonic sections (**f**) from WT and *Rnase4⁻/⁻* mice 8 days after 2.5% DSS administration (*n* = 6). **g, h** Quantitative mRNA expression (**g**) and protein level (**h**) of indicated cytokines in the colons of WT and *Rnase4⁻/⁻* mice 8 days after 2.5% DSS treatment (*n* = 6). Scale bar, 50 μm in H&E staining. Data are presented as mean ± SEM for (**a–d, f** and **h**) and as mean for (**g**); * *p* < 0.05; ** *p* < 0.01; *** *p* < 0.001 by two-tailed unpaired Student's *t*-test (**a–d** and **h**) or two-tailed Mann–Whitney *U* test (**f**).

knockout mice, *Rnase4^ΔIEC* mice exhibited no intestinal abnormalities under normal physiological conditions (Supplementary Fig. 8a–g and Supplementary Fig. 9a–c), except for alterations in the gut microbiota (Supplementary Fig. 10a). Importantly, *Rnase4^ΔIEC* mice also developed severe colitis upon DSS treatment (Supplementary Fig. 10b–g), reinforcing the role of intestinal epithelial cell-derived Rnase4 in modulating colitis susceptibility.

**Rnase4-regulated bacteria play a key role in colitis**

To investigate whether Rnase4 suppresses colitis by modulating gut microorganisms, we designed a microbiota manipulation experiment. WT mice were cohoused with either age- and sex-matched *Rnase4⁻/⁻* mice (WT (co-*Rnase4⁻/⁻*)) or with WT littermates (WT (co-WT)) born from heterozygous *Rnase4⁺/⁻* parents for a period of 6 weeks prior to the induction of colitis (Fig. 4a). The results revealed that WT (co-*Rnase4⁻/⁻*) mice exhibited an exacerbated inflammatory response compared to their non-cohoused WT counterparts or WT (co-WT) controls, as evidenced by colitis indicators (Fig. 4b, c, and Supplementary Fig. 11a–e). The exacerbated colitis observed in WT (co-*Rnase4⁻/⁻*) mice could potentially be attributed to alterations in their gut microbiota composition, resulting from the ingestion of fecal matter from their *Rnase4*-deficient cage mates. To validate this

hypothesis, we performed a fecal microbiota transplantation, where age- and sex-matched WT or *Rnase4⁻/⁻* donor feces were orally gavaged into the gastrointestinal tract of antibiotic-pretreated WT recipient mice every other day for a period of 2 weeks before DSS-induced colitis. The WT recipients that received fecal transplants from *Rnase4⁻/⁻* donors displayed aggravated colitis symptoms relative to those receiving feces from WT donors (Fig. 4d–f, and Supplementary Fig. 11f–k).

To identify the specific colitis-related microbes, we tried to isolate and culture the top enriched bacteria from *Rnase4⁻/⁻* mice, and only successfully obtained a *Parasutterella* strain. The *Mucispirillum* genus, an obligately anaerobic, gram-negative bacterium from the Deferribacterota phylum, has also been reported to be increased with intestinal inflammation and to enhance the development of colitis through degrading mucin[28,29]. However, due to the difficulty in culturing *Mucispirillum* strain under our laboratory conditions, we proceeded with the *Parasutterella* strain for subsequent experiments. To established a robust colonization of the isolated bacterial strain in the murine gut, WT mice were subjected to oral gavage with the bacterial suspension for a period of 2 weeks. Following this inoculation period, the mice were given a 1-week recovery period before the initiation of DSS-induced colitis. The successful establishment of the *Parasutterella*

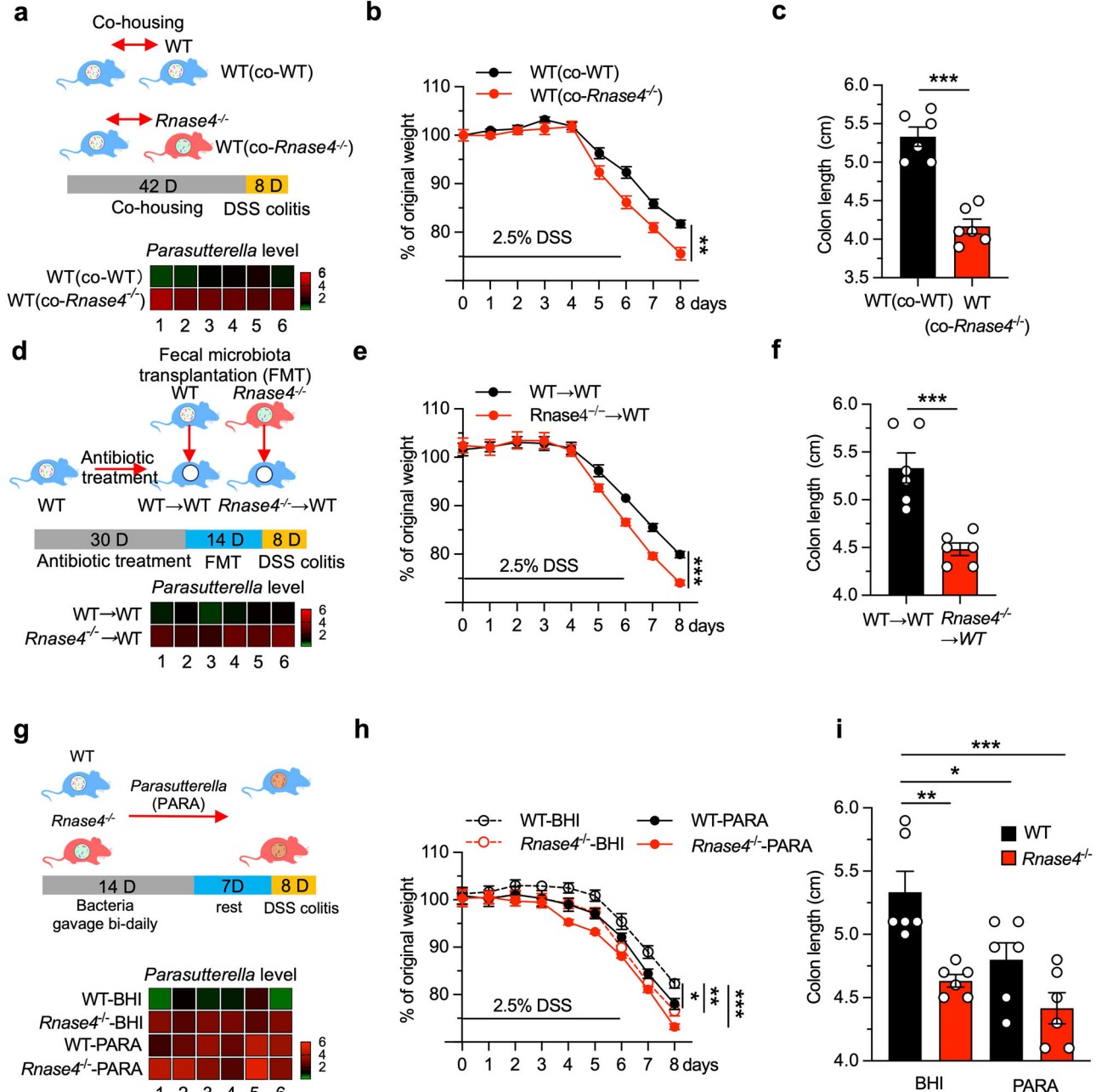

**Fig. 4 | Rnase4-regulated bacteria play a key role in colitis. a** Diagram of the co-housing experiment. The heatmap shows the relative abundance of *Parasutterella* in each mouse after normalization to the average of WT (co-WT) group. **b, c** Body weight loss (**b**) and colon length (**c**) of WT (co-WT) and WT (co-*Rnase4⁻/⁻*) mice with DSS-induced colitis (*n* = 6). **d** Diagram of the gut microbiota transplantation experiment. The heatmap shows the relative abundance of *Parasutterella* in each mouse after normalization to the average of the WT → WT group. **e, f** Body weight loss (**e**) and colon length (**f**) of WT → WT and *Rnase4⁻/⁻* → WT mice with DSS-induced colitis (*n* = 6). **g** Diagram of *Parasutterella* gavage and DSS induction to analyze the relationship between *Parasutterella* and colitis progression. The heatmap shows the relative abundance of *Parasutterella* in each mouse after normalization to the average of the WT–BHI group. **h, i** Body weight loss (**h**) and colon length (**i**) of WT–BHI, WT–PARA, *Rnase4⁻/⁻*–BHI and *Rnase4⁻/⁻*–PARA mice with DSS-induced colitis (*n* = 6). Data are presented as mean ± SEM for (**b, c, e, f, h** and **i**) and as mean for (**a, d** and **g**); * *p* < 0.05; ** *p* < 0.01; *** *p* < 0.001 by two-tailed unpaired Student's *t*-test (**b, c, e, f, h** and **i**).

strain in the murine gastrointestinal tract was verified by 16 S rDNA quantitative PCR (Fig. 4g). There were no significant effects on body weight in mice fed with bacteria or vehicle control (BHI, used for bacterial culture) before DSS treatment (Supplementary Fig. 12a). Interestingly, upon DSS administration, WT mice gavaged with the *Parasutterella* strain suffered from more severe colitis (Fig. 4h, i and Supplementary Fig. 12b–f), confirming the role of Rnase4-regulated bacteria in colitis.

## Rnase4 directly kills *Parasutterella* and maintains the intestinal tryptophan metabolism

To elucidate the mechanism underlying the regulation of Rnase4 on *Parasutterella*, we first examined the effect of Rnase4 on this specific strain. A dose-dependent reduction in *Parasutterella* viability was observed upon exposure to the recombinant Rnase4 protein, with a median lethal concentration (LC$_{50}$) of 0.37 μM (Fig. 5a). Interestingly, K40A, a Rnase4 mutant lacking ribonuclease activity[30], also showed

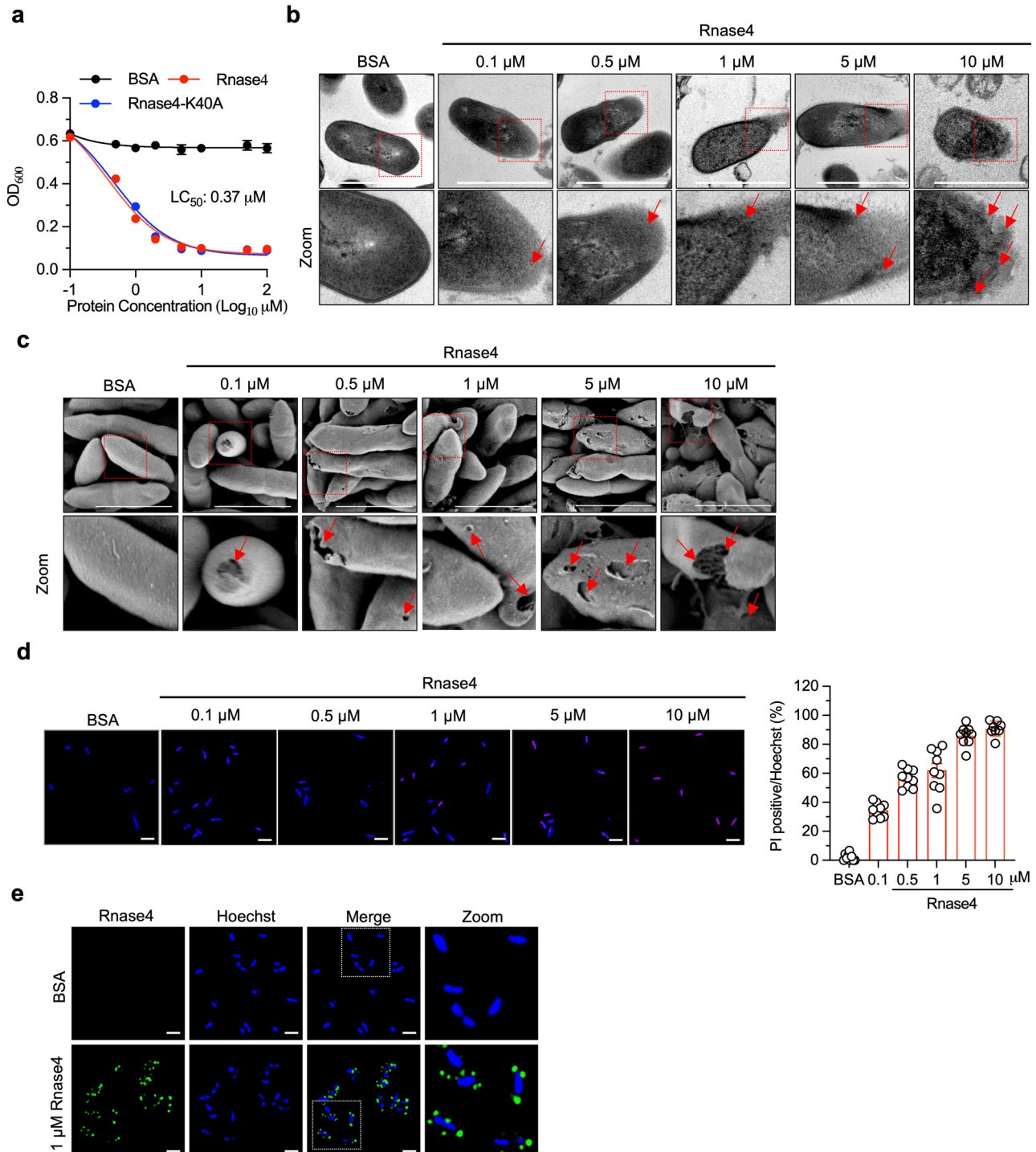

**Fig. 5 | Rnase4 directly kills *Parasutterella*. a** Antibacterial efficacy of recombinant Rnase4 and its enzymatic null mutant Rnase4-K40A against *Parasutterella*. Bovine serum albumin (BSA) served as a control (*n* = 3 independent experiments). **b, c** Transmission (**b**) and scanning (**c**) electron microscopic micrographs of *Parasutterella* treated with different concentrations of Rnase4. The bottom panel of images provides a zoomed-in view of the regions indicated in the top images. The red arrows show the damage to the bacterial cell wall. Scale bar, 1 μm. **d** PI staining of *Parasutterella* treated with different concentrations of Rnase4. The left panel shows the percent of PI positive bacteria in each group (n = 9 fields per group). Scale bar, 1 μm. **e** Immunofluorescence staining of RNASE4 in *Parasutterella* after incubation with Rnase4. Scale bar, 1 μm. Data are presented as mean ± SEM. Representative images from four independent experiments are shown (**b–e**).

such an effect, indicating its bactericidal activity was not related to intrinsic ribonuclease activity. Transmission and scanning electron microscopy revealed that Rnase4 disrupted the *Parasutterella* membranes by forming pores in the bacterial walls, thereby causing cytoplasmic leakage (Fig. 5b, c). This membrane disruption was further supported by the dose-dependent uptake of propidium iodide in *Parasutterella* following Rnase4 treatment, highlighting bacterial membrane permeabilization (Fig. 5d). Moreover, immunofluorescence staining showed that Rnase4 bound to and aggregated on the *Parasutterella* membrane (Fig. 5e), suggesting that its membrane-

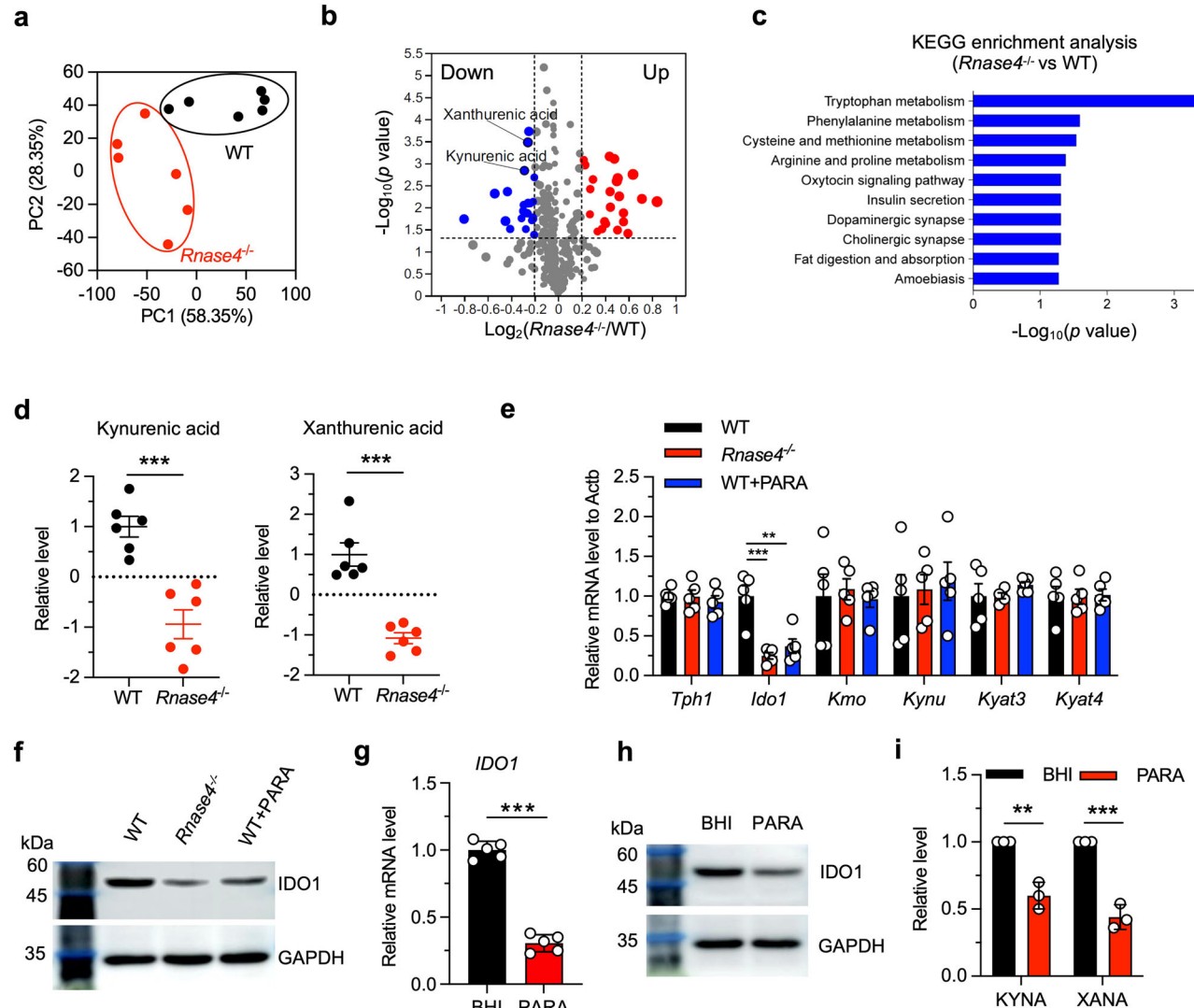

**Fig. 6 | Rnase4 associated *Parasutterella* regulates intestinal tryptophan metabolism. a** Unweighted UniFrac principal-coordinate analysis of the β-diversity of metabolite composition in stool samples of WT or *Rnase4*$^{-/-}$ mice (*n* = 6). **b** Volcano plot showing differentially expressed metabolites between WT and *Rnase4*$^{-/-}$ mice (*n* = 6). **c** KEGG enrichment analysis of differential metabolites between WT and *Rnase4*$^{-/-}$ mice. **d** Relative levels of kynurenic acid and xanthurenic acid in stool samples of WT and *Rnase4*$^{-/-}$ mice (n = 6). **e** Quantitative mRNA expression of selected tryptophan metabolizing genes in colonic tissues from WT, *Rnase4*$^{-/-}$, and *Parasutterella* gavaged mice (WT + PARA) as measured by quantitative PCR (*n* = 5 independent experiments). **f** IDO1 protein levels in colonic tissues from the three groups in (**e**) by immunoblotting. **g–i** Quantitative mRNA (**g**, *n* = 5 independent experiments) and protein expressions of *IDO1* (**h**), and relative levels of kynurenic acid and xanthurenic acid (**i**, *n* = 3 independent experiments) in human intestinal epithelial cells (HIEC-6) treated with *Parasutterella* culture medium. Data are presented as mean ± SEM for (**d, e, g**, and **i**); ** *p* < 0.01; *** *p* < 0.001 by two-tailed unpaired Student's *t*-test (**d, e, g**, and **i**). Significantly altered metabolites were determined using the two-tailed Mann–Whitney *U* test, and adjusted *p* < 0.05 were considered statistically significant (**b**). KEGG enrichment analyses were carried out with the Fisher's exact test, and FDR correction for multiple testing was performed (**c**).

disrupting activity is a result of a direct interaction with the bacterial wall. Collectively, these results reveal that Rnase4 kills *Parasutterella* by disrupting its membrane integrity.

To further explore how *Parasutterella* affects colitis, we analyzed the metabolites in stool samples from WT and *Rnase4*$^{-/-}$ mice. The untargeted metabolomic analysis revealed a significant difference in the metabolic landscape between the two groups (Fig. 6a). In total, 47 metabolites were found to be significantly altered (adjusted *p* < 0.05; Fig. 6b), with a notable enrichment in the tryptophan metabolism pathway (Fig. 6c). Among these metabolites, kynurenic acid (KYNA) and xanthurenic acid (XANA) were significantly decreased in *Rnase4*$^{-/-}$ mice (Fig. 6d). These two metabolites are the end products of tryptophan metabolism, and are known to exhibit protective effects against colitis[8,9]. Interestingly, we found that the expression of *IDO1*, a

rate-limiting enzyme of the tryptophan metabolic pathway[7], was significantly suppressed in the intestinal epithelial cells of the *Rnase4*$^{-/-}$ mice (Fig. 6e, f). We further confirmed that the cultured medium of *Parasutterella* significantly inhibited *IDO1* expression and the production of KYNA and XANA in human intestinal epithelial cell line (HIEC-6) (Fig. 6g–i). These data indicate that *Parasutterella* downregulates tryptophan metabolism in the intestinal epithelium, thus exacerbating colitis.

To confirm the existence of the "Rnase4–*Parasutterella*–tryptophan metabolism–colitis" regulatory axis, we implemented a rescue experiment wherein *Rnase4*$^{-/-}$ mice were orally administered with KYNA and XANA during DSS administration. Remarkably, the treatment effectively alleviated the symptoms of colitis (Supplementary Fig. 13a–g). Overall, our results demonstrated that Rnase4-regulated *Parasutterella*

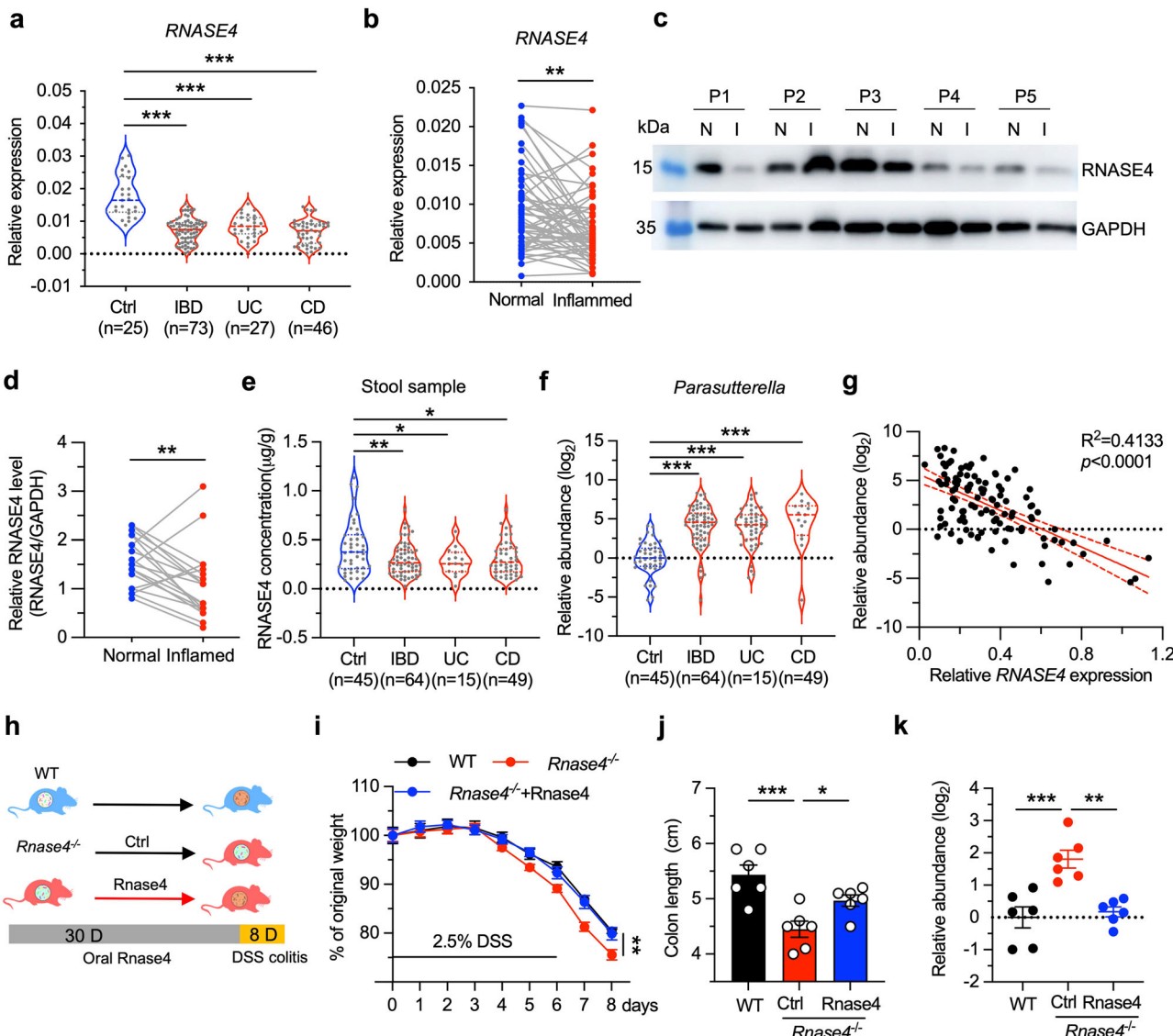

**Fig. 7 | RNASE4 is a potential IBD diagnostic biomarker and therapeutic target.**
**a** Quantitative mRNA expression of *RNASE4* in colon samples from healthy participants (Ctrl, *n* = 25) and patients with IBD (*n* = 73), including ulcerative colitis (*n* = 27) and Crohn's disease (*n* = 46). Expression values were normalized to β-actin. **b** Quantitative mRNA expression of *RNASE4* in normal and inflamed colon samples from patients with IBD (*n* = 40). **c, d** Representative immunoblotting pictures (**c**) and quantitative results (**d**) of RNASE4 protein in normal and inflamed colonic tissues from patients with IBD (*n* = 20). **e, f** RNASE4 concentration (**e**) and relative abundance of *Parasutterella* (**f**) in stool samples from healthy participants (Ctrl, *n* = 45) and patients with IBD (*n* = 64), including UC (*n* = 15) and CD (*n* = 49).

**g** Correlation between RNASE4 level and the abundance of *Parasutterella* in stool samples. **h** Scheme of oral RNASE4 treatment in *Rnase4*⁻/⁻ mice to analyze its preventive potential. **i, j** Body weight loss (**i**) and colon length (**j**) of WT, *Rnase4*⁻/⁻, and *Rnase4*⁻/⁻ supplemented with the recombinant RNASE4 protein (*Rnase4*⁻/⁻+RNASE4) mice during DSS-induced colitis (*n* = 6). **k** Relative abundance of *Parasutterella* in stool samples from WT, *Rnase4*⁻/⁻, and *Rnase4*⁻/⁻ + RNASE4 mice (*n* = 6). Data are presented as mean ± SEM for (**a, e, f, i, j** and **k**); * *p* < 0.05; ** *p* < 0.01; ** *p* < 0.001 by two-tailed paired Student's *t*-test (**b** and **d**), and two-tailed unpaired Student's *t*-test (**a, e, f, i, j** and **k**). The correlation was assessed by simple linear regression analysis (**g**).

downregulated KYNA and XANA metabolism in intestinal epithelial cells, thereby enhancing intestinal susceptibility to colitis.

## RNASE4 is a potential IBD diagnostic biomarker and therapeutic target

The exacerbation of colitis observed in *Rnase4*-deficient mice, as well as the association between Rnase4 and gut microbiota dysbiosis, imply possible clinical potential of RNASE4 in IBD management, including Ulcerative colitis (UC) and Crohn's disease (CD). To explore the diagnostic value of RNASE4, we examined its expression in human intestinal tissue samples, and observed a significant reduction of the mRNA level in patients with IBD (Fig. 7a). In the same patient, *RNASE4* mRNA and protein levels were significantly decreased in inflamed tissue

compared with normal one (Fig. 7b–d). Additionally, we found that the RNASE4 level was 0.30 ± 0.16 μg/g in the stools of patients with IBD, which was approximately 30% lower than in the control group (0.42 ± 0.27 μg/g; Fig. 7e). Conversely, an increased *Parasutterella* level was detected in the stools of patients with IBD (Fig. 7f). These results revealed an inverse correlation between RNASE4 and *Parasutterella* levels (Fig. 7g), suggesting that they may serve as biomarkers for IBD diagnosis. Interestingly, these patterns were not specifically associated with CD or UC, suggesting that RNASE4's role in regulating IBD pathogenesis may represent a general mechanism rather than being linked to a particular disease subtype.

To investigate the therapeutic potential of Rnase4, we orally treated *Rnase4*⁻/⁻ mice with the recombinant Rnase4 protein for 30

days, followed by DSS administration to induce colitis (Fig. 7h). We first examined the levels of Rnase4 in the small intestinal lumen and stool samples of the treated mice. The concentrations of Rnase4 in the small intestinal lumen and stool of Rnase4-treated mice were 0.62 μM and 0.44 μM, respectively, which are comparable to the levels found in WT mice and close to the $LC_{50}$ of Rnase4 against *Parasutterella* (Supplementary Fig. 14a, b). These findings suggested that exogenous supplementation of Rnase4 could achieve physiologically relevant levels. Notably, mice pretreated with Rnase4 exhibited attenuated signs of colitis (Fig. 7i, j and Supplementary Fig. 15a–e). The amelioration of disease symptoms was accompanied by a significant reduction in the level of *Parasutterella* (Fig. 7k), implying that reversion of dysbiosis underlies the therapeutic efficacy of Rnase4 in experimental colitis, at least partially. This finding highlighted the possible clinical application of RNASE4 as a novel treatment strategy for IBD.

## Discussion

As the most evolutionarily conserved gene in vertebrate[31], Rnase4 exhibits antimicrobial activity against uropathogenic *Escherichia coli* (UPEC) and multidrug-resistant UPEC in urinary tract[32–34]. Here, we identify Rnase4 as a new intestinal antimicrobial protein, functioning to limit the growth of *Parasutterella* and maintain gut microbiota balance. We further elucidate the antibacterial mechanism of this protein on *Parasutterella*, and establish a host-microbiota interaction pathway—i.e., intestinally secreted Rnase4–gut *Parasutterella*–epithelium tryptophan metabolism–gut microbiota—that contributes to the pathogenesis of colitis. Importantly, we observe a decreased level of RNASE4 and increased abundance of *Parasutterella* in IBD patient samples, and demonstrate that exogenous Rnase4 supplementation can restore the gut microbiome composition and attenuate the severity of colitis. These results highlight the importance of Rnase4 in maintaining intestinal microbiota, and suggest potential clinical applications in IBD.

As a matter of fact, a diverse array of antimicrobial peptides and proteins exists in the intestinal lumen and works in concert to maintain microbiota homeostasis and protect the intestinal epithelium[35]. These include α-defensins, which are expressed exclusively in Paneth cells[10,11,36], and others such as cathelicidins[14], REG3γ[15,16], and ANG[37]. Abnormal production of these molecules is associated with various disorders, such as IBD, obesity, and liver diseases[18]. For instance, a reduction in α-defensins alters gut microbiota composition, increases bacterial translocation across the epithelial barrier, resulting in excessive immune activation and chronic intestinal inflammation[12]. As a newly identified intestinal antimicrobial protein, although currently we do not know when Rnase4 would be expressed and its underlying regulation mechanism, our data reveal that its deficiency does not change the expression levels of major known intestinal antimicrobial peptides and proteins. Considering its evolutionary conservation, we suspect that Rnase4 might be a core antimicrobial protein and serve as a complementary or synergistic factor to the existing arsenal of antimicrobial molecules. Meanwhile, our results indicate that the absence of *Rnase4* does not significantly impact intestinal structure, cell composition, morphology and cell polarity of Paneth and goblet cells as well. However, the transmission electron microscopy analysis showed that the granules in Paneth cells of *Rnase4*−/− mice appeared to have more vacuoles compared to those in WT mice (Supplementary Fig. 4b). This phenomenon suggests that Rnase4 may influence the formation and maturation process of secretory vesicles. It has been reported that there are abnormalities in granules of Paneth cells in patients with Crohn's disease and in the mouse models[38,39]. These abnormalities are thought to reflect or be caused by disturbances in endoplasmic reticulum function and autophagy pathways[40,41]. Therefore, it would be warranted to analyze the changes of key molecules related to ER stress or autophagy in response to *Rnase4* deletion in future studies. That would help to fully elucidate the underlying mechanisms of IBD.

Generally, antimicrobial proteins exert their antimicrobial effects via the following mechanisms: agglutination of the pathogens[42]; translocation across the membrane and possible interaction with intracellular targets[43]; total solubilization of the membrane[44]; or formation of pores in the membrane[45]. We found that Rnase4 created pore-like structures in the wall of *Parasutterella*, suggesting that it kills the bacteria by forming membrane pores. Pore-forming antimicrobial proteins typically possess characteristics such as amphiphilicity and a positive net charge[46], which are also present in Rnase4. Currently, there are two established pore-formation models: barrel-stave and toroidal. In both models, the hydrophobic residues of antimicrobial proteins interact with the lipid bilayer in the membrane core, while the hydrophilic residues form a polar channel. Barrel-stave pores are compact, bundle-like assemblies of antimicrobial proteins with little effect on the neighboring lipids[47]. By contrast, antimicrobial proteins in toroidal pores are loosely arranged, and the lipid headgroups are present in the polar channel[48]. Due to the small size and transient nature of the pores, determining the structure of Rnase4-induced pores in *Parasutterella* is challenging. In the future, modern technologies—such as super-resolution microscopy and cryo-EM tomography—may enable the capture of these transient structures, and provide a molecular-level understanding of the antimicrobial activity of Rnase4.

Among the Rnase4-targeting bacteria, *Parasutterella* has been reported to be associated with various health outcomes, including CD, irritable bowel syndrome, obesity, and type 2 diabetes[5,6,49]. For example, Ju et al. demonstrated that *Parasutterella* colonization in mice significantly altered the cecal metabolome without shifting the microbial structure, particularly affecting the kynurenine pathway of tryptophan metabolism[4], which is consistent with our observation. However, the mechanism underlying these changes has not been investigated yet. Our study provides valuable insight into how *Parasutterella* promotes colitis; i.e., by suppressing host *IDO1* expression and subsequently altering host tryptophan metabolism. Coincidentally, *Parasutterella* itself lacks key tryptophan metabolism-related genes in its genome, supporting our hypothesis that its regulation of the host tryptophan metabolism occurs through a paracrine manner. In other words, *Parasutterella* may secrete factors that influence *IDO1* expression in host epithelial cells, rather than directly metabolizing tryptophan by itself. Here we would like to propose that the regulatory role of Rnase4 on the core gut microbiota is a fundamental mechanism of host-microbiota interaction which has been preserved throughout the evolution. Further studies should focus on the underlying cellular and molecular mechanisms mediating the interactions between core bacterial strains and the host.

In this study, we observed decreased RNASE4 levels, increased *Parasutterella* abundance, and downregulated tryptophan metabolism in both the experimental colitis model and patients with IBD, suggesting that these factors might serve as IBD diagnostic biomarkers. Traditionally, the diagnosis of IBD is mainly achieved via expert assessment based on clinical symptoms, combined with results obtained from endoscopic and radiological examinations[50]. However, this approach is labor-intensive and time-consuming, and there is an increasing need to develop body fluid detection methods. Nowadays, fecal calprotectin has emerged as a widely used biomarker for monitoring disease activity and predicting relapse in IBD patients[51]. Similarly, cytokines such as IL-6, TNF-α, and IL-1β have been shown to play a crucial role in the pathogenesis of IBD, and their levels correlate with disease severity[52]. Additionally, the abundance of certain intestinal microbiota, such as *Ruminococcus gnavus*[53], *Prevotella copri*[54], has been used to assess the progression of IBD[55]. Therefore, it would be interesting to test the diagnostic value of various combinations of these factors in a large cohort, with a hope to develop precision IBD diagnosis techniques at the earliest stage possible, thus enabling earlier intervention and more personalized treatment[56].

Our data also suggest that Rnase4 may serve as an intervention agent. As a matter of fact, epidemiological studies have shown that higher consumption of dairy products and milk is associated with a lower risk of developing IBD[57], and Rnase4 has long been known to be present in milk with antimicrobial activity[58]. Combined with our findings, these observations indicate that Rnase4 can protect the bowel from damage by modulating the microbiota, and could be developed as a functional additive or even a therapeutic drug. Interestingly, Laura et al. have identified histone deacetylase inhibitors as a class of drugs that enhance *RNASE4* expression from a high-throughput screen of medications approved by the United States Food and Drug Administration (FDA)[34]. Therefore, enhancement of endogenous *RNASE4* expression and/or activity may be another therapeutic strategy. Certainly, these insights into the potential therapeutic applications of RNASE4 warrant further investigation.

In conclusion, our work provides a valuable example of the intricate interplay between the host and gut microbiota, highlighting the significant role of Rnase4 in maintaining microbial balance and preventing intestinal inflammation.

## Methods

### Human samples
The human samples used in this study were collected and described in our previous work[27,59]. The study included two sample cohorts, with patient characteristics detailed in Supplementary Table 2. The first cohort primarily consisted of intestinal tissue samples, which were used for RNA and protein analyses. The second cohort mainly comprised stool samples, which were utilized for investigating fecal protein levels and gut microbiota composition. Briefly, intestinal tissues and stool samples were obtained from the Inflammatory Bowel Disease Center at the Sir Run Run Shaw Hospital, affiliated with the Zhejiang University School of Medicine. The samples were immediately frozen at −20 °C within 2 h of collection, and subsequently transported to the laboratory on dry ice, where they were stored at −80 °C for further analysis. The diagnosis of Crohn's disease and ulcerative colitis was based on a standard combination of clinical, endoscopic, histological, and radiological criteria. Ethical approval for the use of human samples was obtained from the ethics committee of Sir Run Run Shaw Hospital, Zhejiang University School of Medicine (#20210622-31); waiver for informed written consent was also approved simultaneously, as the deidentified biopsies/samples were all previously collected.

### Mice
Transcription activator-like effector nuclease (TALEN)-based *Rnase4* knockout mice (*Rnase4$^{-/-}$*) and CRISPR-Cas9-based *Rnase4* conditional knockout mice (*Rnase4$^{fl/fl}$*) were generated in collaboration with Cyagen Biosciences. For the TALEN-based knockout strategy, the *Rnase4* exon 2 region (GenBank accession number: NM_201239.3) located on mouse chromosome 14: 51,104,841-51,104,895 (GRCm38/mm10) was selected as the TALEN target site (Supplementary Fig. 2a). TALENs were constructed using the Golden Gate Assembly and confirmed by sequencing. TALEN mRNA, generated by in vitro transcription, was injected into fertilized eggs for knockout mice production. The founder (F0) mice, which showed an inserting of 2 bases (TG) in both strands compared to the wild-type DNA sequence, were genotyped by PCR combined with DNA-sequencing analysis. Heterozygous F1 mice were produced by mating F0 with WT mice. Heterozygous F1 mice were then mated to produce F2 mice. Homozygote F2 mice (*Rnase4$^{-/-}$*) were determined by DNA-sequencing analysis (Supplementary Fig. 2b) and for subsequent assays.

For CRISPR-Cas9 mediated gene targeting strategy, the *Rnase4* exon 2 region was selected and a floxed allele was constructed. Two guide RNAs (gRNAs) were designed to bind to 399 base pairs (bp) upstream and 627 bp downstream of exon 2. A cocktail of two gRNAs, donor vector containing loxP sites flanking the targeted exon along with 5′ and 3′ homology arms, and Cas9 nuclease mRNA were co-injected into fertilized mouse eggs for knockin mice production. The founder (F0) mice, which contained two loxP sites in upstream and downstream of exon 2, were confirmed by PCR and DNA-sequencing analysis. Heterozygous F1 mice were generated by mating F0 with WT mice. These heterozygous F1 mice were then bred to produce F2 mice. Homozygote F2 mice (*Rnase4$^{fl/fl}$*) were identified by PCR (Supplementary Fig. 7b) and used for subsequent assays.

For mouse breeding, *Rnase4$^{-/-}$* mice and their WT controls were generated from the same heterozygous *Rnase4$^{+/-}$* parents, while *Rnase4$^{\Delta IEC}$* mice and their controls (*Rnase4$^{fl/fl}$*) were generated from *Rnase4$^{fl/fl}$* mice and Villin-cre mice (Cre recombinase was expressed in villus and crypt epithelial cells of the small and large intestines, #T000142, GemPharmatech). The genotypes were subsequently confirmed by Sanger sequencing and polymerase chain reaction (PCR) amplification; the genotyping primers are listed in Supplementary Table 3. Littermates were randomly assigned to experimental groups. All mice were maintained in specific pathogen-free (SPF) conditions (temperatures of -18–23 °C with 40–60% humidity) with a standard 12-h daylight cycle at the Laboratory Animal Centre of Zhejiang University, and provided with water and a standard laboratory diet ad libitum, except where otherwise noted. All animal studies were performed in compliance with the guide for the care and use of laboratory animals, adopting the protocol approved by the Medical Experimental Animal Care Commission of Zhejiang University (#ZJU20220219).

### Cell line
The human intestinal epithelial cell line HIEC-6 was purchased from American Type Culture Collection (#CRL-3266) and cultured in Opti-MEM (#31985070; Gibco) supplemented with 10% fetal bovine serum (FBS, #26010074; Thermo Fisher Scientific) and 1% (v/v) penicillin-streptomycin (#15140122; Thermo Fisher Scientific) at 37 °C under 5% CO$_2$. To investigate the effect of *Parasutterella* on cell metabolism, HIEC-6 cells were treated with the culture medium of *Parasutterella* or brain heart infusion (BHI, #237300; Thermo Fisher Scientific) for 4 h. Subsequently, the expression of *IDO1* and the production of kynurenic acid (KYNA) and xanthurenic acid (XANA) were analyzed.

### Bacterial strain isolation and identification
The top enriched bacteria were isolated from fecal samples of *Rnase4$^{-/-}$* mice via the following procedure (Supplementary Fig. 16): first, two fresh fecal pellets were collected and homogenized in 1 ml of BHI broth containing 5% FBS, 0.1% cysteine (#168149; Sigma-Aldrich), and 4 μg/ml oxacillin (#A600677; Sangon biotech). Serial dilutions of the homogenates were plated onto pre-reduced, anaerobically sterilized BHI plates supplemented with 5% FBS and 0.1% cysteine. After anaerobic incubation at 37 °C for 72 h, individual colonies were streaked onto fresh BHI plates and incubated for an additional 48 h. A single colony was then selected and grown in BHI broth for 24 h. The genomic DNA was extracted from the culture, and a segment of the 16 S rRNA gene was amplified using the primers detailed in Supplementary Table 4. The amplified gene segments were then sequenced, and the taxonomy of each strain was identified via a Basic Local Alignment Search Tool (BLAST, https://blast.ncbi.nlm.nih.gov).

### Rnase4 expression and purification
Mouse recombinant Rnase4 and ribonuclease-inactive variant (Rnase4-K40A) proteins were generated with a pET *Escherichia coli* expression system, and purified by SP-Sepharose and reverse-phase high-performance liquid chromatography, as previously described[60]. The purity and molecular weight of the recombinant proteins were assessed by SDS-PAGE analysis (Supplementary Fig. 17c).

Ribonucleolytic activities of recombinant Rnase4 and its variant K40A were examined using total RNA and yeast tRNA as the substrate. For the total RNA cleavage assay, 1 μg of each protein was added to a

final reaction volume of 300 µl, containing 1 µg HeLa cell total RNA in a reaction buffer (0.33 M Hepes, 0.33 M NaCl, pH 7.0, and 0.1 mg/ml RNASE-free BSA). After incubation at 37 °C for 30 min, 10 µl of each reaction sample were analyzed by RNA agarose gel electrophoresis to assess the integrity of the total RNA (Supplementary Fig. 17d). For yeast tRNA cleavage assay, varying amount proteins were added to a final volume of 300 µl, containing 600 ng of yeast tRNA (#AM7119; Thermo Fisher Scientific) in the same reaction buffer as described above. After incubation at 37 °C for 120 min, 700 µl of ice-cold 3.4 % perchloric acid (#244252; Sigma-Aldrich) was added and incubated on ice for 10 min, centrifuge at 14,000×$g$ for 10 min at 4 °C, and the absorbance at 260 nm of the supernatants was recorded (Supplementary Fig. 17e). All buffers and water used above were prepared by Ultrapure DNase/RNase-Free distilled water (#10977015; Thermo Fisher Scientific) to ensure that the system was RNASE-free.

### Electron microscopy (TEM) analysis

Mouse intestine segments (0.5 cm in length) from eight-week-old C57BL/6 J WT male mouse were fixed in 4% glutaraldehyde solution (#G7776; Sigma-Aldrich), followed by post-fixation in 1% osmium tetroxide (#201030; Sigma-Aldrich). The samples were then dehydrated in a graded series of ethanol (#459844; Sigma-Aldrich) and propylene oxide (#110205; Sigma-Aldrich), and embedded in resin (#45359; Sigma-Aldrich). Ultrathin sections were cut using a LEICA EM UC7 cryo-ultramicrotome, stained with 2% uranyl acetate (#21447, Polysciences), and observed under a Thermo Scientific Talos L120C 120 kV transmission electron microscope.

### Mouse intestinal organoid culture

Small intestine from eight-week-old C57BL/6 J WT male mouse was dissected, opened longitudinally, and briefly washed with ice-cold Dulbecco's phosphate-buffered saline (DPBS, #14190144, Thermo Fisher Scientific). The tissue was further dissected into small pieces, and the villi were removed by scraping with a glass coverslip. The intestinal fragments were then washed with cold DPBS to remove unattached epithelial fragments. To isolate the crypts, the intestinal fragments were incubated in DPBS containing 3 mM EDTA (#AM9912; Thermo Fisher Scientific) for 15 min on ice. After removing the EDTA solution, the fragments were washed once with DPBS and then vigorously suspended in DPBS using a 10 mL pipette. The supernatant containing the villous fraction was discarded, and the sediment was resuspended in PBS. After further vigorous suspension, the supernatant, which was enriched for crypts, was collected and passed through a 70 µm cell strainer (for small intestine) or 100 µm cell strainer (for colon) to remove residual villous material. The filtered suspension was then centrifuged at 80–100 g for 3 min to separate crypts from single cells. The purified crypts were counted and used for organoid culture. Typically, 50–200 crypts were embedded in 20 µL Matrigel (#356231; BD Biosciences) and cultured with IntestiCult Organoid Growth Medium (#06005; STEM CELL Technologies) supplemented with 100 U/mL penicillin and 100 µg/mL streptomycin (#15070063; Thermo Fisher Scientific). The culture medium was changed every 2–3 days, and the organoids were incubated under standard tissue culture conditions (37 °C, 5% $CO_2$). Subsequently, the organoids were embedded, sectioned, and used for staining.

### Flow cytometry for sorting intestinal epithelial cells

A single crypt cell suspension was filtered through a 40 µm cell strainer and simultaneously labelled with the following fluorescence-conjugated antibodies in staining buffer (2 mM EDTA (#AM9260G; Thermo Fisher Scientific) and 3% FBS (#10100147; Thermo Fisher Scientific) in PBS): CD31-PE, CD45-PE, EpCAM-APC and CD24-PerCP-Cyanine 5.5. For sorting colonic goblet cells, the cells labeled with the following antibodies: CD31-BV 510, CD45-BV 510, EpCAM-eFluor™ 450 and UEA I-DyLight 649 (the antibody information detailed in

Supplementary Table 7). After gently shaking for 30 min on ice, the cells were washed with staining buffer three times, resuspended at a concentration of $5 \times 10^6$ cells/mL in Advanced DMEM/F12 medium (#12634010; Thermo Fisher Scientific) supplemented with 10 µg/mL DAPI (to distinguish live cells from dead/dying cells, #564907; BD Biosciences), and analyzed with an LSR Fortessa flow cytometer. Cell population analysis was performed with FlowJo software v10.4. The markers for intestinal stem cells were Lgr5-eGFP$^{hi}$; EpCAM$^+$; CD24$^{med/-}$; CD31$^-$; CD45$^-$; DAPI$^-$; transit-amplifying cells were Lgr5-eGFP$^{low}$; EpCAM$^+$; CD24$^{med/-}$; CD31$^-$; CD45$^-$; DAPI$^-$; Paneth cells were Lgr5-eGFP$^{neg}$; EpCAM$^+$; CD24$^{hi}$; CD31$^-$; CD45$^-$; DAPI$^-$; Side scatter$^{hi}$; and colonic goblet cells were UEA I$^{hi}$; EpCAM$^+$; CD31$^-$; CD45$^-$; DAPI$^-$ (Supplementary Fig. 18).

### Experimental colitis

To establish dextran sulfate sodium (DSS)-induced colitis model, eight-week-old male mice were treated with DSS (2.5% w/vol, molecular weight: 36–50 kDa, #160110; MP Biomedicals) in drinking water for 6 days, followed by regular drinking water. To establish 2,4,6-trinitrobenzene sulfonic acid solution (TNBS)-induced colitis model, 8-week-old male mice were anesthetized, and 100 µL of TNBS solution (1 volume of 5% w/vol TNBS solution mixed with 1 volume of absolute ethanol, #P2297; Sigma-Aldrich) was slowly instilled into the lumen of the colon. After treatment, the mice were euthanized using carbon dioxide, and colonic tissues were collected for further examination.

Disease activity index scores were used to evaluate the clinical signs of mouse colitis, including body weight loss, occult blood, and stool consistency[61]. Mice were blindly scored to evaluate the colitis phenotype. The weight loss score was graded as follows: 0, no weight loss; 1, loss of 1–5% original weight; 2, loss of 6–10% original weight; 3, loss of 11–20% original weight; 4, loss of >20% original weight. The bleeding score was determined as follows: 0, no blood determined by Hemoccult (Beckman Coulter) analysis; 1, positive Hemocult; 2, visible blood traces in stool; 3, gross rectal bleeding. The stool score was determined as follows: 0, well-formed pellets; 1, semiformed stool that did not adhere to the anus; 2, pasty, semiformed stool that adhered to the anus; 3, liquid stool that adhered to the anus.

### Cohousing experiment

Three-week-old and sex-matched male wild-type (WT) or $Rnase4^{-/-}$ mice, derived from the same $Rnase4^{+/-}$ breeding pair, were cohoused with three-week-old C57BL/6 J WT mice in a new cage at a 1:1 ratio for 6 weeks (WT (co-WT) or WT (co-$Rnase4^{-/-}$)). Fecal samples were collected at the end of the cohousing experiment, and the abundance of target bacteria was analyzed using quantitative PCR.

### Fecal microbiota transplantation

Fecal microbiota transplantation was conducted in 8-week-old C57BL/6 J WT male mice, who were administered an antibiotic cocktail (1 g/L ampicillin (#A9518; Sigma-Aldrich), 0.5 g/L vancomycin (#V2002; Sigma-Aldrich), 1 g/L neomycin (#N6386; Sigma-Aldrich), 1 g/L metronidazole (#M1547; Sigma-Aldrich) in sterilized water) for 30 days to eliminate resident microbiota. Ablation of resident microbiota was confirmed by aerobic/anaerobic plating on BHI plates supplemented with 5% FBS. The antibiotic cocktail was discontinued 24 h prior to transplantation. Fecal microbiota transplant donors were weight-, age-, and sex-matched male WT and $Rnase4^{-/-}$ littermates. Fresh fecal pellets were homogenized in sterile PBS (100 mg/ml), and fecal suspensions were immediately gavaged into recipient mice (100 µl/mouse) to minimize air exposure. Mice were transplanted every other day for 2 weeks, and then subjected to DSS administration.

### Mouse colonization with specific bacteria

The bacterial strain was cultured as described below and centrifuged to obtain a bacterial pellet, which was then resuspended in prereduced

BHI at an approximate concentration of $10^9$ colony-forming units (CFU)/ml. Both four-week-old WT and *Rnase4*$^{-/-}$ male mice were orally administered 200 μl of *Parasutterella* suspension every other day for two weeks, while the vehicle BHI was gavaged as a control. Following the final gavage, the colonization efficiency was determined via quantitative PCR, after which the mice were challenged with DSS.

## Fluorescein isothiocyanate (FITC) labeled dextran permeability assay

Intestinal permeability was evaluated via the oral administration of FITC labeled dextran, as previously described[62]. In brief, the mice fasted for 4 h and were then intragastrically treated with FITC-dextran tracer (4000 Da, 0.6 mg/g body weight, #FD150S; Sigma-Aldrich) in 100 μl PBS. After 4 h, mouse blood was collected to obtain the hemolysis-free serum for fluorescence intensity detection using the Thermo Scientific™ Varioskan Flash fluorescence spectrophotometer (excitation: 488 nm, emission: 520 nm). A standard curve for FITC−dextran was prepared by serially diluting a known amount of FITC−dextran in PBS; it was then used to calculate the concentration of FITC−dextran in mouse serum.

## Mouse oral treatment with Rnase4

To investigate the effect of Rnase4 treatment on colitis, four-week-old *Rnase4*$^{-/-}$ male mice were administered recombinant Rnase4 protein (20 mg/kg per mouse) in drinking water for 30 days. Thereafter, they were subjected to DSS treatment, and the clinical symptoms of colitis and fecal bacterial composition were analyzed.

## 16 S rDNA high-throughput sequencing and analysis

Total fecal bacterial DNA was extracted using the QIAamp DNA Stool Mini Kit (#51604; Qiagen) according to the manufacturer's instruction, with the addition of a bead-beating step to increase yield. The V3−V4 hypervariable region of the 16 S rDNA was amplified using a universal forward sequencing primer, and a uniquely barcoded reverse sequencing primer to allow for multiplexing. The amplified DNA was then sequenced using Realbio technology[63].

The sequence reads were analyzed using the QIIME (quantitative insights into microbial ecology) analysis pipeline as previously described[64]. Briefly, the pipeline used FASTA quality files and a mapping file indicating the barcoded sequence corresponding with each sample, as inputs. Reads were split by sample according to the barcode, taxonomical classification was performed using the RDP-classifier, and an operational taxonomic unit (OTU) table was created. Closed reference OTU mapping was employed using the RDP database (http://rdp.cme.msu.edu). Sequences sharing 97% nucleotide sequence identity in the V3−V4 region were binned into OTUs (97% ID OTUs). The QIIME analysis pipeline was used to perform α-diversity (observed species) and β-diversity analysis (unweighted UniFrac distance). Differential species associated with particular interventions were identified via the linear discriminant analysis effect size, with an effect size threshold of 2.

## Fecal metabolomics analysis

Metabolite extraction from stool, nontargeted liquid chromatography-tandem mass spectrometry (LC-MS/MS) analysis, and data pre-processing and annotation were performed by Majorbio Bio-Pharm Technology. Briefly, 100 mg of stool sample was used for the ultra-high performance liquid chromatography-quadrupole time-of-flight mass spectrometry (UHPLC/Q-TOF-MS) analysis. Nontargeted LC-MS/MS analysis was performed using the 1290 UHPLC system (Agilent Technologies) with a UPLC HSS T3 column (1.8 μm, 2.1*100 mm; Waters Corporation) coupled to the TripleTOF 5600 (Q-TOF; AB Sciex). The Triple TOF mass spectrometer was used to acquire MS/MS spectra on an information-dependent basis during the LC/MS experiment.

MS raw data files were converted to mzXML format using ProteoWizard, and processed using the R package XCMS (version 3.2). The preprocessing results generated a data matrix that included retention time, mass-to-charge ratio (m/z) values, and peak intensity. R package CAMERA was used for peak annotation after XCMS data processing. An MS2 database was applied for metabolite identification.

The metabolomic data was analyzed using R package MetaboAnalystR. Significantly altered metabolites were determined using the two-tailed Mann−Whitney $U$ test, and adjusted $p < 0.05$ were considered statistically significant.

## Microbicidal activity assay

To evaluate antimicrobial activity, bacterial growth was assessed by measuring the optical density at 600 nm ($OD_{600}$). Specifically, bacteria were grown to midlogarithmic phase in an appropriate medium; this was followed by centrifugation at $3000g$ for 5 min. The pellet was resuspended and further diluted to a final concentration of $10^6$ CFU/ml in 10 mM sodium phosphate buffer (pH 7.2). In total, 50 μl of the bacterial suspension was incubated with 50 μl of recombinant Rnase4, Rnase4-K40A, or BSA at varying concentrations for 2 h at 37 °C. Growth inhibition was assessed by inoculating the bacterial suspension into 96-well plates and incubating at 37 °C for 48 h. The $OD_{600}$ was measured and used to plot growth curves. The median lethal dose ($LD_{50}$) was determined as the protein concentration resulting in 50% inhibition of bacterial growth.

## Scanning and transmission electron microscopy

Bacteria harvested during the midlogarithmic growth phase were pelleted by centrifugation at $3000g$ for 5 min. The pellet was then resuspended and diluted to an $OD_{600}$ of 0.1 in 10 mM sodium phosphate buffer (pH 7.2). The bacteria were anaerobically incubated with Rnase4 at different concentrations for 2 h at 37 °C. Subsequently, the bacteria were fixed in PBS containing 2.5% glutaraldehyde (#A600875; Sangon biotech) for 12 h at 4 °C. For scanning electron microscopy, the bacteria sample was rinsed three times with PBS buffer, and postfixed in 1% osmium tetroxide (#20816-12-0; Sigma-Aldrich) for 1 h. The sample was then examined using an FEI Tencai Biotwin scanning electron microscope. For transmission electron microscopy, the bacterial sample was rinsed three times with PBS buffer, postfixed in 2% uranium acetate (#6159-44-0; SPI Supplies) for 30 min, and examined using a Tecnai G2 spirit 120 kV transmission electron microscope.

## Hematoxylin and eosin (H&E)

The mouse intestinal tract was obtained immediately after carbon dioxide euthanasia, longitudinally opened, and washed with 70% ethanol. It was then fixed in a 10% formalin solution overnight, embedded in paraffin, and sliced into sections. Deparaffinization was performed by incubating the sections in xylene (#534056; Sigma-Aldrich) for 30 min, and rehydrating in decreasing ethanol solutions (100%, 95%, and 80%) for 5 min each, and then distilled water for 5 min. H&E staining was performed by immersing the sections in hematoxylin (#C0107; Beyotime) for 5 min, followed by PBS for 3 min (to prevent background staining), eosin (#C0109; Beyotime) for 2 min, and finally, distilled water for 5 min. The stained sections were mounted using neutral resin and a coverslip, and observed and digitally photographed using a Leica DM2000 LED microscope.

## Histological analysis

A blinded, board-certified pathologist histologically assessed colitis based on the previously described criteria[46], including the extent and severity of inflammation, and ulceration of the mucosa. The severity score for inflammation was as follows: 0, normal (within the normal limit); 1, mild (small, focal, or widely separated, limited to the lamina propria); 2, moderate (multifocal or locally extensive, extending to the

submucosa); and 3, severe (transmural inflammation with ulcers covering >20 crypts). The score for ulceration was as follows: 0, normal (no ulcers); 1, mild (1–2 ulcers involving up to a total of 20 crypts); 2, moderate (3–4 ulcers involving a total of 20–40 crypts); and 3, severe (>4 ulcers or >40 crypts).

## Periodic acid schiff (PAS) staining

The mouse intestinal sections were deparaffinized with xylene for 10 min, rehydrated using decreasing concentrations of ethanol (100%, 95%, and 80%) for 3 min each, and incubated in distilled water for 3 min. Subsequently, the sections were treated with an iodic acid solution for 6 min at room temperature using PAS kit (#C0142, Beyotime). Thereafter, they were incubated with Schiff reagent at room temperature in the dark for 15 min, and rinsed with distilled water for 10 min. The sections were then incubated with hematoxylin staining solution for 2 min, and sealed after rinsing with tap water for 15 min. The stained sections were observed and digitally photographed using a Leica DM2000 LED microscope.

## Immunofluorescence staining

The intestinal sections were first deparaffinized in xylene for 10 min, rehydrated in descending ethanol solutions (100%, 95%, and 80%) for 3 min each, and incubated in distilled water for 3 min. High-temperature antigen retrieval in 0.01 M citrate buffer (pH 6.0; #E673002; Sangon Biotech) was performed for 20 min, followed by blocking in 10% normal goat serum (#E510009; Sangon Biotech) with 0.3% Triton-X 100 for 30 min at room temperature. The sections were then incubated with primary antibodies overnight at 4 °C, including mouse anti-Ki67, rabbit anti-Lysozyme, and rabbit anti-Rnase4. For secondary labeling, the sections were incubated with donkey anti-rabbit IgG conjugated to Alexa Fluor 488/555, or anti-mouse IgG conjugated to Alexa Fluor 488 for 1 h at 37 °C. Nuclei were stained with Hoechst 33342 (10 µg/mL, #H3570; Thermo Fisher Scientific) for 30 min at room temperature, and the mucus layer was stained with WGA-FITC (1:1000; #W11261; Thermo Fisher Scientific) for 30 min at room temperature. Fluorescence images were acquired at randomly selected locations using a Nikon A1 confocal microscope system. The antibody information detailed in Supplementary Table 7.

## Fluorescence in situ hybridization (FISH)

A section of the mouse distal colon was preserved in a 4% paraformaldehyde solution at 4 °C overnight. The colon tissue was then transferred through 15% and 30% sucrose solutions, and embedded in optimal cutting temperature compound (#4583, Tissue-Tek); longitudinal sections (5 µm) were cut using a cryostat (Leica). The slides were treated with hybridization buffer (0.9 M NaCl, 20 mM Tris-HCl, 0.01% sodium dodecyl sulfate, 10% formamide; pH 7.5) and incubated for 14 h at 42 °C in a humidified chamber with 10 ng/µl FISH probe (Genscript); the probe information is listed in Supplementary Table 5. Subsequently, the slides were incubated in wash buffer (0.9 M NaCl, 20 mM Tris-HCl; pH 7.5) preheated to 42 °C for 20 min, and washed gently three times. The samples were stained with 10 µg/ml Hoechst 33342 in PBS for 15 min at room temperature in the dark, washed three times with PBS, and finally mounted in Vectashield mounting medium (#H-1900; Vector Labs). The images were captured using a Nikon A1 confocal microscope.

## Real-time quantitative PCR analysis

For gut bacteria analysis, bacterial DNA was extracted from feces using the QIAamp DNA Stool Mini Kit (#51604; Qiagen); the quantity and quality of extracted DNA were determined using a NanoDrop 2000 spectrophotometer (Thermo Fisher Scientific). A 10-ng template of extracted DNA was used for quantitative PCR analysis, and the primer sets used are listed in Supplementary Table 4.

For cytokine gene quantification, total RNAs from the colonic tissue of DSS-treated mice were purified via precipitation with lithium chloride as previously reported[65]. The mRNA level was evaluated using a reverse transcription reaction with Moloney Murine Leukemia Virus (M-MLV) reverse transcriptase (#639574; Takara), followed by quantitative PCR analysis with SYBR Premix Ex Taq (#RR420A; Takara) on a Roche 480 real-time PCR system. The primer sets used for quantitative PCR are listed in Supplementary Table 6.

## Immunoblot and coomassie blue staining

RIPA buffer (#89900; Thermo Fisher Scientific) supplemented with a protease inhibitor cocktail (#11697498001; Roche) was used to extract proteins from colon, mucus layer, and stool samples. The protein concentration was measured using a BCA protein assay kit (#23225; Thermo Fisher Scientific). The resolved proteins were then subjected to 15% sodium dodecyl sulfate-polyacrylamide gel electrophoresis. For Coomassie Blue Staining, the gel was impregnated with Coomassie brilliant blue R250 (#ST031; Beyotime) for 1 h, and then decolorized with eluent (#P0017C, Beyotime) overnight.

For immunoblotting, the separated proteins were transferred onto a 0.45 µm polyvinylidene fluoride membrane (#88518; Thermo Fisher Scientific), or the protein solution was directly spotted onto the PVDF membrane. The membrane was then blocked with 5% BSA in Tris-buffered saline containing 0.05% Tween 20 (TBST) for 1 h. The membrane was washed in TBST three times and subsequently incubated with primary antibodies diluted in 5% BSA on a rocker at 4 °C overnight. After washing with TBST another three times, the membrane was incubated with HRP-conjugated secondary antibody (1:1000; #31430 or #31460; Thermo Fisher Scientific) diluted in 5% BSA for 1 h at room temperature. The chemiluminescence signals were detected using SuperSignal West Pico Chemiluminescent Substrate (#34580; Thermo Fisher Scientific), and the images were captured using the GE Healthcare Life Sciences Amersham Imager 600. The primary antibodies were detailed in Supplementary Table 7.

## Fecal RNASE4 measurement

The stool samples were thawed at room temperature, and 100 mg of stool was homogenized in 0.5 ml of extraction buffer (pH 8.0), containing Tris 0.1 mM, citric acid 0.1 mM, urea 1.0 mM, $CaCl_2$ 0.01 mM, and protease inhibitor cocktail. The samples were vigorously mixed for 30 min and centrifuged at 15,000$g$ for 20 min to collect the supernatant. The supernatant was filtered using a 5 µm cutoff filter, and the total protein concentration was estimated using a BCA protein assay kit. The concentration of fecal RNASE4 was quantitatively measured using the enzyme-linked immunosorbent assay (ELISA) technique as previously described[60]. Each sample was run with a blank, standard, and control. The RNASE4 level was expressed as µg/g per sample of total protein.

## Quantitative kynurenic acid (KYNA) and xanthurenic acid (XANA) level

The levels of KYNA and XANA were analyzed using ultra-high-performance liquid chromatography-tandem mass spectrometry (UHPLC-MS/MS). Briefly, 100 µL of the sample was combined with 10 µL of an internal standard solution (Trp-D5, 4000 ng/mL; #615862; Sigma-Aldrich), and then added to 990 µL of extraction solution (acetonitrile: water = 9:1; #34851; Sigma-Aldrich). The solution was then vortexed for 30 s and sonicated at 40 kHz for 30 min at 5 °C. After sonication, the sample was stored at −20 °C for 30 min to facilitate the precipitation of proteins and other particulate matter. Finally, the sample was centrifuged at 13,000$g$ for 15 min at 4 °C, and the supernatant was collected for analysis.

The UHPLC-MS/MS analysis was performed using a Nexera Series LC-40 system coupled with a QTRAP® 6500+ mass spectrometer (Sciex) at Majorbio Bio-Pharm Technology. Chromatographic

separation was achieved using an ACQUITY UPLC HSS T3 thermostatted column (2.1 × 150 mm, 1.8 μm) set to 40 °C, with a mobile phase comprising water containing 0.1% formic acid (solvent A), and acetonitrile in water containing 0.1% formic acid (solvent B). The gradient elution conditions were as follows: 0.0–2.5 min, 1–11% B; 2.5–5.5 min, maintain 11% B; 5.5–6.5 min, 11–28% B; 6.5–7.5 min, maintain 28% B; 7.5–12.5 min, 28–50% B; 12.5–13.5 min, 50–95% B; 13.5–15.5 min, maintain 95% B; 15.5–15.6 min, 95–1% B; 15.6–18 min, maintain 1% B. The total chromatographic separation time was 18 min with a flow rate of 1 mL/min. The samples were stored at 4 °C during the analysis.

The MS data were acquired using a QTRAP 6500+ mass spectrometer (Sciex) equipped with an electrospray ionization source operating in both positive and negative mode. The source temperature was set to 550 °C, curtain gas was maintained at 35 psi, charged aerosol detector gas pressure was set to medium, and both ion source gas 1 and 2 were set to 50 psi. Ion-spray voltage floating was set to 5500 V/−4500 V.

The LC-MS raw data were analyzed using Sciex software OS, with ion fragments automatically identified and integrated using default parameters. Manual inspection was performed to confirm the quality of the data, and the metabolite concentrations of the samples were determined by comparing the peak areas of the analytes with those of the internal standards, using a linear regression standard curve.

### Sample size and sample collection

No statistical methods were used to predetermine sample size for in vivo and in vitro experiments, but at least three samples were used per experimental group and condition. The number of samples used in each experiment is indicated in the Source Data. Samples and experimental animals were randomly assigned to experimental groups. Animal procedures (that is, genotyping and treatments) were performed by investigators unaware of the experimental design.

### Statistical analysis

Data were presented as the mean with the standard error of the mean (SEM), and all heatmaps represent the mean values. Normality was checked using the Shapiro–Wilk test. For comparisons between two groups, two-tailed unpaired or paired Student's $t$-tests were used for normally distributed data, and Mann–Whitney $U$ tests were used for non-normally distributed data. The significant separation of the microbiome composition was assessed by analysis of similarities test. The correlation between RNASE4 level and the abundance of *Parasutterella* in stool samples was assessed by simple linear regression analysis. All statistical analyses were performed using GraphPad Prism v10. The following significance levels were used: * $p < 0.05$; ** $p < 0.01$; *** $p < 0.001$.

### Reporting summary

Further information on research design is available in the Nature Portfolio Reporting Summary linked to this article.

## Data availability

The raw 16 s rDNA sequencing data have been deposited in NCBI's Sequence Read Archive (SRA) under BioProject accession number PRJNA1008371. Source data are provided with this paper.

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

## Acknowledgements

This study was supported by the following grants: National Natural Science Foundation of China (No. U21A20202 to Z.X., No. U23A20448 to Z.X., No. 81972612 to Z.X., No. 32071289 to J.Sh.); Natural Science Foundation of Zhejiang Province (2021R51002 to Z.X.); Fundamental Research Funds for the Central Universities (No. 2021QNA7009 to J.Sh.); Leading Innovative and Entrepreneur Team Introduction Program of Zhejiang (No. 2021R01012 to R.B.); Leading Innovation and Entrepreneur Team of Hangzhou (No. TD2020006 to R.B.). We thank Prof. Wei Liu, Prof. Qiming Sun, and Prof. Di Wang (Zhejiang University School of Medicine) for critical suggestion to the project conceptualization and experimental design; Yanwei Li, Shuangshuang Liu, Guifeng Xiao, Guizhen Zhu, and Chenyu Yang from the Core Facilities (Zhejiang University School of Medicine) for technical assistance in protein purification, microscopy analysis, TEM and SEM analysis.

## Author contributions

J.S., M.C., Z.H. and N.X. share co-first authorship. J.S., J.Sh., R.B. and Z.X. conceived of the project. J.S., M.C., Z.H., W.W., N.X. and Z.P. performed the mouse experiments. J.S. and D.S. performed bacterial 16 S rDNA high-throughput sequencing analysis. J.S. conducted the metabolomic analysis. H.Z. and J.Z. performed the molecular and cellular experiments. J.S., Z.Z. and H.L. were involved in the bacteriology experiments. X.G., L.L. and W.Z. provided essential materials and analyzed the clinical samples. J.S. and J.Sh. wrote the original draft. J.Sh., R.B. and Z.X. reviewed and revised the final version of the text. J.Sh. and Z.X. supervised the study.

## Competing interests

The authors declare no competing interests.
