## [Peer Review File · Nature Communications]

Ribonuclease 4 functions as an intestinal antimicrobial protein to maintain gut microbiota and metabolite homeostasisREVIEWER COMMENTS

Reviewer #1 (Mucosal immunity) (Remarks to the Author):

In this paper, Sun et al study the role of Ribonuclease 4 as an antimicrobial factor that regulates the intestinal microbiota and influences the local tissue response to inflammation. They show that lack of RNASE4 in epithelial cells leads to overgrowth of certain microbial species including *Parasutterella* and that this organism enhances susceptibility to colitis via an ability to downregulate IDO1 mediated production of kynurenic and xanthurenic acid. The work is presented clearly and the data generally support the conclusions made. By linking a specific component of the microbiota to intestinal inflammation and determining an underlying mechanism, the findings are novel and may be of translational relevance in human IBD. Some comments to be considered:

- 1) It is likely that *Parasutterella* will not be that familiar to many readers and therefore it would be appropriate for the authors to provide more information about this organism, what is known about its possible involvement in regulating immune responses, disease susceptibility etc.
- 2) This would also be useful in helping to explain why the authors selected this organism from amongst what appears to be a large number of changes in the composition of the microbiota in RNASE4^{-/-} mice. For instance, no information is provided as to which of the various classes/phyla/orders/families etc shown in Extended Figure 4 *Parasutterella* might belong to. From these figures, it would appear that as well as the *Mucispirillum* genus discussed, several other groups of bacteria are just as different in RNASE4^{-/-} mice as the *Parasutterella*. For instance, *Muribaculaceae* spp seem to be the most over-represented, but are not discussed, while the *Parasutterella* and *Mucispirillum* seem to be minor components. More justification of the choice of *Parasutterella* is required.
- 3) The staining for *Parasutterella* and *Mucispirillum* shown in Figure 2 is very weak and difficult to distinguish from background.
- 4) The ability of *Parasutterella* to modify disease susceptibility when used as a single species is interesting. However some comment would be warranted on how this can have a significant effect when added to the established microbiota of a WT mouse.

Reviewer #2 (Microbiota, IBD) (Remarks to the Author):

In the manuscript "Ribonuclease 4 functions as an intestinal antimicrobial protein to maintain gut homeostasis" the authors report decreased RNASE4 levels in patients with IBD as well as increased *Parasutterella* levels in IBD stool samples. RNASE4 is an intestinal antimicrobial protein with unclear function. In this paper the authors described RNASE4 as being highly expressed in intestinal epithelium where it seems to particularly target *Parasutterella* strains. RNASE4 knockout mice as well as epithelium specific RNASE4 knockout are more susceptible to experimental colitis. Additionally the authors show that *Parasutterella* down regulate the expression of an rate limiting enzyme of the tryptophan metabolism in intestinal epithelial cells, which led to the author's conclusion that RNASE4 is important for gut homeostasis and might function as a diagnostic biomarker for IBD.

The study indicates that RNASE4 kills commensal gut bacteria like *Parasutterella* in mice, likely by pore formation in bacterial membranes, which does not require the ribonuclease activity. Deletion of this antimicrobial increases susceptibility to experimental gut inflammation, which may involve tryptophan metabolism of gut epithelium through IDO1.

Application of RNASE4 ameliorates experimental colitis. As such the authors identify disease relevant antimicrobial in the gut with functions similar to other RNASEs that could become relevant in IBD.

The following points should be addressed:

An unbiased survey of RNASE4 actions on gut commensals is lacking. Can the authors better delineate the effect on RNASE4 on the gut microbiota? For example, the authors show convincingly the overgrowth of two bacterial genera but only work with the *Parasutterella* genus. Was the second genus also tested?

Are WT mice exposed to *parasutterella* susceptible to DSS colitis? This would prove the inflammatory actions of *Parasutterella* independent from RNASE4 activity.

Is RNASE4 expression in stool associated with specific IBD features? The patient cohort is not explained and any relation to RNASE4 is not explained.

How specific is RNASE4 expression to human Paneth cells or goblet cells? Can the authors analyse human sc seq data to show the distribution – which would be relevant for future biomarker studies in IBD.

Which *Parasutterella* spp are cultured? And how do the authors grow top enriched bacteria in RNASE4 mice? This is not explained or shown and appears little convincing.

Regarding the mouse models; is there any more information on the *Parasutterella* strain?

Was the viability of the inoculated bacteria tested after the administration? qPCR of the stool can not distinguish between dead or viable bacteria.

The bactericidal effect of RNASE4 on *Parasutterella* was nicely shown, but is this *Parasutterella* specific? How is the effect of RNASE4 on other species, especially on *Mucispirillum*? Is this effect known for other RNASE?

Have *Rnase4*^{-/-} mice microbial phenotypes beyond the colonic gut microbiota? Also in the small intestine and skin or is it relevant only in the colon?

Was the RNASE4 knockout mouse also tested in other facility? Can the author rule out that this effect might be due to specific housing (and a yet undefined pathobiont)?

Reviewer #3 (Defensin/AMP, microbiota, IBD) (Remarks to the Author):

The authors of the manuscript hypothesized for the first time that RNASE4 regulates the gut microbiota as an antimicrobial protein and showed that RNASE4 expressed in certain gut epithelial cells including goblet cell and Paneth cell have some effects on shaping the gut microbiota as well as a protective role in DSS colitis via *Parasutterella*. The authors also claimed that RNASE4 is an antimicrobial protein that maintains microbial balance and prevents inflammation in the gut by a series of mouse experiments and analyzing samples from IBD patients, further may serve as intervention agent for IBD patients. The concept of the study itself is quite new and interesting in the related field. However, there are a series of concerns especially in the study design, materials, interpretation of results, and discussion. Many of those are critical. Major issues are shown below along with Minor issues.

Major:

#1 The authors should define antimicrobial peptide (AMP) following current general understandings of those. Despite the authors described that RNASE4 is an antimicrobial protein but not a peptide, there are contamination in use of the term AMP everywhere in the manuscript.

In addition, it is inappropriate to refer to the products/secretions of the two cells, goblet cell and Paneth cell, together as AMPs. To date, AMPs in mammalian small intestinal epithelial cells especially mice and humans are secreted almost exclusively from Paneth cells, but not

goblet cells and the others. More importantly, goblet cells secrete mucins that harbor the gut microbiota, but not AMPs. All the inadequate terms and sentences should be corrected.

#2 The authors showed that both goblet and Paneth cells are immunostained with RNASE4 and described that RNASE4 is secreted from both cells to contribute immunity. However, the issue important here is that we don't know all the effects shown in the manuscript came from which cell or even the other cell. Where are the fecal RNASE4 coming from? The authors only showed the immunostainings of RNASE4, then, described that feces and colonic mucosal residues contain RNASE4 in some level. However, the authors failed to provide clear evidence that it is secreted from goblet cells and/or Paneth cells.

In addition, if these cells secrete RNASE4, a number of important questions related to the underlying mechanism raises. For example, how the cells or cell secrete RNASE4? What kind of stimuli let them secrete? At this point, the authors failed to provide the underlying mechanism for observed results in the manuscript.

Thus, to support their conclusions, carefully designed additional experiments such as cultured intestinal organoids and/or other methods would be needed. Note that so many cell lineages in the intestine express RNASE4.

#3 In this study, two genera, *Mucospirillum* and *Parasutterella* were most enriched in RNASE4 ^{-/-}, in addition to decreasing diversity. Then, the increased *Parasutterella* only was highlighted as dysbiosis related with the DSS colitis and IBD. The statements about dysbiosis in both Results and Discussion in the manuscript, especially in relation to experimental colitis and IBD remind us some critical questions. Whether the dysbiosis has links to the patients with UC or CD and how these bacteria contribute to the pathology in the colon and ileum, etc. The authors need to explain such connections if any. Most importantly, much detail and precise evaluation of the gut microbiota is mandatory.

#4 The authors showed killing effects of RNASE4 only against *Parasutterella*, but not other commensal species in the gut. Does RNASE4 have bactericidal activities against gut pathogenic bacteria such as Gram-negative *E. coli* and Gram-positives *S. aureus* as well as other major commensal bacteria such as *Lactobacillus*, *Clostridium*, and *Bacteroidetes*? To understand the gut microbiota in their ecosystem, not just focus on *Parasutterella* but answering these questions is very important, so that further give insights in the underlying mechanism how RNASE4 shapes the composition of gut microbes.

#5 The authors of the manuscript showed the involvement of RNASE4 in shaping the gut microbiota but did not measure the levels of the major enteric AMPs that function in the gut lumen reported in previous references. That recalls a question whether abnormalities in α -defensins, cryptdins and HD5, and other AMPs such as Reg3 γ and angiogenin are responsible for the pathology shown even partially. Since AMPs including α -defensins, cryptdins, HD5, Reg3 γ , angiogenin, etc., secreted from Paneth cells have been known to contribute to the gut innate immunity and regulation of the gut microbiota community, the authors should include these points in Discussion and add levels of those in the RNASE4^{-/-} as a new result. To exclude the contribution of these representative AMPs to the observed dysbiosis in RNASE4^{-/-}, it is interesting to see levels of such representative AMPs in feces of RNASE4^{-/-} mice.

#6 Related with #2, whether the observed intestinal microbiota changes, dysbiosis, was due to abnormalities in Paneth cells or goblet cells or anything else? Because no phenotypic changes were found in both cells and even entire intestine in RNASE4^{-/-} mice, the authors tried to explain all the effects seen by the intervention, owing to observed molecular changes

in some intestinal epithelial cells. However, a vital question why the dysbiosis in RNASE4^{-/-} both without and with intervention occurred remain unknown or too short to be proved, so that we have to say the underlying mechanisms of the effects in RNASE4^{-/-} mice is puzzle.

Minor:

#7 Although the recombinant RNASE4 and the RNASE4 extracted from mouse intestine were used in this study, no molecular information such as amino acids and structure were provided in Materials and Methods section. So, it is difficult to evaluate qualities of those RNASE4. Need to add detailed characteristics of the RNASE4 used in the manuscript.

#8 In the manuscript, the authors just stated that no phenotype was found in RNASE4^{-/-} mice on the intestinal epithelial cells including goblet and Paneth cells. How far did the authors investigate before reaching that conclusion? It is important to know whether the structure of the intestinal epithelium, especially the morphology of secretory granules and cell polarity are also normal. Since the granule shape itself has been reported to be very important especially for Paneth cells in terms of functions including Paneth cells in CD, the authors should conduct appropriate analyses to confirm this issue.

#9 It was reported that in angiogenin^{-/-} mice, Lachnospiraceae decreased while α -Proteobacteria increased. It is interesting to know similar changes happened in RNASE4^{-/-} mice.

#10 Page 8 and Fig 4e: It is hard to find positive meanings on that fecal RNASE4 protein expression in IBD patients was 1.4-fold lower than control subjects. What do the authors try to say by that?

#11 Page 11: Methods in diagnosis of IBD has been largely developing these days. The sentences in Discussion section about IBD diagnosis was not quite right, neglecting recent references such as calprotectin and cytokines.

#12 The authors administered RNASE4 orally for the treatment of DSS colitis in the manuscript and Extended Data Fig.15 but did not show that administered RNASE4 was gone through the intestinal lumen and elicit the effects. Did the authors check RNASE4 concentration in the intestinal lumen or the feces during RNASE4 administration? It should be added because this could be critical data to support their conclusion.

#13 The authors showed mRNA expression of several potentially functional molecules including cytokines. How about the expression of protein levels of those molecules?

#14 It would be interesting to see what happened in the systemic immunity in RNASE4^{-/-} mice? Is the dysbiosis and the susceptibility to DSS colitis occurred in RNASE4^{-/-} mice simply because leaky gut? Or are there any further effects in innate and adaptive immunity?

#15 Discussion and References: The authors missed many important previous studies to discuss and cite. Discussion section is insufficient and need to reconstruct, partially due to lack of previous important research results and logical configuration. References shown in the manuscript are inadequate and many important studies are missing, so it should be thoroughly checked and revised.

#16 Most Figures of histology with HE staining and immunohistochemistry need to replace to

clear images or at least add photos with high magnification views, because the important parts affecting credibility of the results are not clear.

Point-by-point Response to Reviewers' Comments

We thank the reviewers for their constructive remarks on our manuscript. We have taken the comments on board to improve and clarify the manuscript. Please find below a detailed point-by-point response to all comments (please note that we have labeled the reviewers' comments in black and our replies in blue, and organized the supporting data as Figure 1, Figure 2, etc.).

Reviewer #1 (Mucosal immunity)

General comments:

In this paper, Sun et al study the role of Ribonuclease 4 as an antimicrobial factor that regulates the intestinal microbiota and influences the local tissue response to inflammation. They show that lack of RNASE4 in epithelial cells leads to overgrowth of certain microbial species including *Parasutterella* and that this organism enhances susceptibility to colitis via an ability to downregulate IDO1 mediated production of kynurenic and xanthurenic acid. The work is presented clearly and the data generally support the conclusions made. By linking a specific component of the microbiota to intestinal inflammation and determining an underlying mechanism, the findings are novel and may be of translational relevance in human IBD.

Specific comments:

- 1) It is likely that *Parasutterella* will not be that familiar to many readers and therefore it would be appropriate for the authors to provide more information about this organism, what is known about its possible involvement in regulating immune responses, disease susceptibility etc.

Response: Thank you for your friendly warning. We agree that background information about *Parasutterella* would benefit the readers to better understand our work. Therefore, we have made the following revisions: 1) we have included the taxonomic information of *Parasutterella* in Supplementary Table 1 to provide an overview of its classification; 2) we have integrated the current understanding about *Parasutterella* into the DISCUSSION section to better contextualize our findings (please refer to line 249-255 of the revised manuscript).

2) This would also be useful in helping to explain why the authors selected this organism from amongst what appears to be a large number of changes in the composition of the microbiota in *RNASE4^{-/-}* mice. For instance, no information is provided as to which of the various classes/phyla/orders/families etc shown in Extended Figure 4 *Parasutterella* might belong to. From these figures, it would appear that as well as the *Mucispirillum* genus discussed, several other groups of bacteria are just as different in *RNASE4^{-/-}* mice as the *Parasutterella*. For instance, Muribaculaceae spp seem to be the most over-represented, but are not discussed, while the *Parasutterella* and *Mucispirillum* seem to be minor components. More justification of the choice of *Parasutterella* is required.

Response: Thank you for raising this important point. First, we would like to clarify that the results presented in Extended Fig. 4 of the previous manuscript (now Supplementary Fig. 5 in this revised manuscript) are based on relative abundance analysis. These results provide an overview of the differences in bacterial composition between the two groups but do not account for within-group variations. For example, although *Muribaculaceae spp.* appear to have a higher mean abundance in *Rnase4^{-/-}* mice, the difference is not statistically significant due to large within-group variations, as shown in **Figure 1** of this response. Second, we want to emphasize that linear discriminant analysis effect size (LEfSe) has comprehensively considered statistical significance, biological consistency, and effect size and supplemented this information in revised manuscript for better understanding. Therefore, based on the significance of the differential bacteria and the LEfSe analysis results, we concluded that *Parasutterella* and *Mucispirillum* were significantly enriched in *Rnase4^{-/-}* mice compared to WT mice.

As for the selection of *Parasutterella*, the only reason is we could not culture *Mucispirillum* in our laboratory.

We have integrated these information into our revised manuscript (please refer to line 98-113 and 148-152 of the revised manuscript).

Figure 1. The relative abundance of bacterial genera in WT and *Rnase4^{-/-}* mice

3) The staining for *Parasutterella* and *Mucispirillum* shown in Figure 2 is very weak and difficult to distinguish from background.

Response: Thank you for pointing out the staining issue for *Parasutterella* and *Mucispirillum* in Fig. 2f. We apologize for the lack of clarity in the original images. To address this issue, we have adjusted the contrast of the FISH staining results in Fig. 2f of the revised manuscript (Figure 2), making the signals more distinct and easier to distinguish from the background.

Figure 2 Fluorescent *in situ* hybridization of colonic lumen sections

4) The ability of *Parasutterella* to modify disease susceptibility when used as a single species is interesting. However, some comment would be warranted on how this can have a significant effect when added to the established microbiota of a WT mouse.

Response: Thank you for raising this interesting point. Although we do not fully know why a single bacterium can significantly change the disease susceptibility, our data clearly showed that the introduction of *Parasutterella* alone significantly promoted WT mouse susceptibility to DSS treatment. We further provided evidence to show that *Parasutterella* inhibited *IDO1* expression in the intestinal

epithelium and downregulated tryptophan metabolism, which may partly explain its significant effect in WT mice. Actually, *Ju et al.* also found that *Parasutterella* can alter the cecal metabolome without significantly shifting the overall microbial structure (*ISME J*, 2019;13(6):1520). To address your concern, we have expanded the discussion in the revised manuscript to provide our mechanistic explanation (please refer to line 256-264 of the revised manuscript).

Reviewer #2 (Microbiota, IBD)

General comments:

In the manuscript “Ribonuclease 4 functions as an intestinal antimicrobial protein to maintain gut homeostasis” the authors report decreased RNASE4 levels in patients with IBD as well as increased *Parasutterella* levels in IBD stool samples. RNASE4 is an intestinal antimicrobial protein with unclear function. In this paper the authors described RNASE4 as being highly expressed in intestinal epithelium where it seems to particularly target *Parasutterella* strains. *RNASE4* knockout mice as well as epithelium specific *RNASE4* knockout are more susceptible to experimental colitis. Additionally, the authors show that *Parasutterella* down regulate the expression of a rate limiting enzyme of the tryptophan metabolism in intestinal epithelial cells, which led to the author’s conclusion that RNASE4 is important for gut homeostasis and might function as a diagnostic biomarker for IBD.

The study indicates that RNASE4 kills commensal gut bacteria like *Parasutterella* in mice, likely by pore formation in bacterial membranes, which does not require the ribonuclease activity. Deletion of this antimicrobial increases susceptibility to experimental gut inflammation, which may involve tryptophan metabolism of gut epithelium through IDO1. Application of RNASE4 ameliorates experimental colitis. As such the authors identify disease relevant antimicrobial in the gut with functions similar to other RNASEs that could become relevant in IBD.

Specific comments:

- 1) An unbiased survey of RNASE4 actions on gut commensals is lacking. Can the authors better delineate the effect on RNASE4 on the gut microbiota? For example, the authors show convincingly the overgrowth of two bacterial genera but only work with the *Parasutterella* genus. Was the second genus also tested?

Response: We appreciate your request to provide an unbiased survey of RNASE4 actions on gut commensals. To address your concern, we have now included a more comprehensive and systematic description of our results, including the differences in bacterial taxa at various taxonomic levels, and have marked these differences in the Supplementary Fig. 5 of the revised manuscript. We hope that these efforts will provide a clearer overview of the impact of RNASE4 on gut microbiota.

As to your second concern, we did attempt to isolate *Mucispirillum* from both mouse and human fecal samples. Unfortunately, under our current experimental conditions, we were unable to expand the culture, and thus could not test its function in the mouse model. We have now included the PCR results from our isolated single-clone of this bacterium to confirm its presence (Supplementary Fig. 16 of the revised manuscript). We have also provided the current understanding on this genus in the RESULTS section (please refer to line 148-152 of the revised manuscript).

2) Are WT mice exposed to *Parasutterella* susceptible to DSS colitis? this would prove the inflammatory actions of *Parasutterella* independent from RNASE4 activity.

Response: Thank you for your question. To determine the function of *Parasutterella* in intestines, we did investigate the colitis susceptibility of WT mice after exposure to *Parasutterella*. As shown in **Figure 3** (Fig. 4g-i of the revised manuscript), WT mice colonized with *Parasutterella* exhibited increased susceptibility to DSS-induced colitis compared to controls, demonstrating that *Parasutterella* can exacerbate intestinal inflammation independent of the RNASE4 activity.

Figure 3 *Parasutterella* can exacerbate intestinal inflammation in WT mice

One may then question the antibacterial ability of RNASE4 in the intestines. Yes, WT mice have normal RNASE4 function, but its level of RNASE4 is relatively low compared to the exogenously

supplemented *Parasutterella*. Therefore, at least a portion of the introduced *Parasutterella* may not be killed by RNASE4 in WT mice, and thus contributes to the observed pro-inflammatory effect.

3) Is RNASE4 expression in stool associated with specific IBD features? The patient cohort is not explained and any relation to RNASE4 is not explained.

Response: Thank you for raising these two important questions.

To clarify the potential relationship between RNASE4 expression and specific IBD features, we have now categorized the IBD patients into CD and UC subgroups, and found that RNASE4 levels in stool were significantly reduced in both subgroups, while *Parasutterella* levels were significantly increased in both groups (Figure 4, Fig. 7e-f of the revised manuscript), indicating that their levels are not linked to a particular disease subtype.

Figure 4 RNASE4 concentration (e) and relative abundance of *Parasutterella* (f) in stool samples

Regarding the patient cohort used in this study, we apologize for not providing a detailed description in the previous manuscript. The clinical samples used for our stool-based analyses were primarily derived from the same patient cohort as our earlier research (*Gut*, 2021;70(4):666), which we cited but did not describe in detail. To overcome this oversight, we have now updated the human subject information with comprehensive information about the patient cohort, including the number of patients, demographic data, and disease characteristics (see the Supplementary Table 2).

4) How specific is RNASE4 expression to human Paneth cells or goblet cells? Can the authors analyze human sc seq data to show the distribution – which would be relevant for future biomarker studies in IBD.

Response: We highly appreciate your constructive suggestion. Following your instruction, we have analyzed the published single-cell sequencing data from the Human Protein Atlas database (www.proteinatlas.org) with 31 different datasets. These datasets were sourced from various platforms including the Single Cell Expression Atlas, the Human Cell Atlas, the Gene Expression Omnibus, the Allen Brain Map, the European Genome-phenome Archive, and the Tabula Sapiens. The results were largely consistent with single-cell sequencing findings in mice, showing that RNASE4 is expressed in various intestinal epithelial cells, with the highest expression observed in goblet cells and Paneth cells. The human single-cell expression data for RNASE4 has now been incorporated into Fig.1b of the revised manuscript (**Figure 5**).

Figure 5 *Rnase4* expression levels in different intestinal cell types based on single-cell RNA sequencing data, sourced from Mouse Cell Atlas (bis.zju.edu.cn/MCA, a) and Human Protein Atlas (www.proteinatlas.org, b)

5) Which *Parasutterella* spp are cultured? And how do the authors grow top enriched bacteria in RNASE4 mice? This is not explained or shown and appears little convincing.

Response: Thank you for your friendly warning. The isolated *Parasutterella* strain was identified as *Parasutterella excrementihominis*, which represents one of the most common subtypes of this species. The bacterial isolation and identification procedure is now provided in the revised METHODS section (please refer to line 340-350 of the revised manuscript, **Figure 6**).

Figure 6 Bacterial strain isolation and identification procedure.

a Workflow for isolating and identifying bacterial strains from the fecal samples of *Rnase4^{-/-}* mice. **b-c** Images of agarose gel electrophoresis show the 16S rDNA PCR results for *Parasutterella* (**b**) or *Mucispirillum* (**c**) positive colony. The PCR amplicon is sequenced, and a BLAST search is performed to identify the bacterial strain.

6) Regarding the mouse models; is there any more information on the *Parasutterella* strain? Was the viability of the inoculated bacteria tested after the administration? qPCR of the stool can not distinguish between dead or viable bacteria.

Response: Thank you for raising this important point regarding microbiota transplantation. Currently, qPCR technology indeed cannot distinguish the dead and viable bacteria. However, the following lines of evidence support that we have successfully colonized *Parasutterella* in the gut:

- 1) As shown in Fig. 4g of the manuscript, seven days post-gavage, we detected a significant increase of *Parasutterella* in the feces of mice compared to the non-gavaged control group. This indicates that *Parasutterella* was present in the intestinal tract after administration. If the bacteria were not viable, their levels would have likely returned to baseline due to elimination through fecal excretion.
- 2) Compared to the control group, we found that the transplantation of *Parasutterella* indeed exacerbated intestinal inflammation, indicating its biological activity and potential impact on gut health (Fig. 4h-I of the manuscript). This suggests that the introduced *Parasutterella* remained functionally active within the gut environment.
- 3) To further address your concern, we isolated feces from the mice post-gavage and cultured them *in vitro* for a week, and observed a proliferative trend of *Parasutterella* (**Figure 7**), suggesting that the bacteria are viable and able to grow.

Taken together, these points suggest that *Parasutterella* not only successfully colonizes in the gut post-transplantation but also remains viable and functionally active, contributing to the observed effects in the mouse models.

Figure 7 Fecal microbiota culture and *Parasutterella* examination

a Schematic diagram of the experimental procedure. **b** *Parasutterella*-positive clones in plate after 7 days culture. **c**, *Parasutterella* abundance at different culture time points.

7) The bactericidal effect of RNASE4 on *Parasutterella* was nicely shown, but is this *Parasutterella* specific? How is the effect of RNASE4 on other species, especially on *mucispirillum*? Is this effect known for other RNASE?

Response: Thank you for your question. Regrettably, we have not yet been able to isolate *Mucispirillum*, and therefore could not carry out bactericidal experiments on this species. However, we have analyzed the effect of RNASE4 on several common Gram-negative and Gram-positive bacteria. As shown in **Figure 8**, the results indicate that RNASE4 has a significantly higher inhibitory efficiency on Gram-negative bacteria compared to Gram-positive bacteria. Among the bacteria we tested, RNASE4 exhibited a notably higher inhibitory effect on *Parasutterella*, with a steeper S-shaped inhibition curve and lower LC₅₀.

Regarding the comparison of RNASE4 activity with other family members, our recent publication titled "*An Evaluation of the Antimicrobial Activity of the Ribonuclease A Superfamily*" (*Chinese Journal of Biochemistry and Molecular Biology*, 2023;(4):524-530) compared the antimicrobial activities of 8 family members against Gram-negative and Gram-positive bacteria. The results showed that although family members tend to preferentially inhibit Gram-negative bacteria, their activities against different bacteria vary significantly. For example, our previous paper (*Gut*, 2021;70(4):666) revealed that angiogenin, another member of this family, significantly inhibits the growth of Gram-negative bacteria such as *Brevundimonas diminuta* (LC₅₀: 0.8 μM) and *Sphingomonas paucimobilis* (LC₅₀: 0.6 μM), but the inhibitory effect of RNASE4 on these two bacteria (LC₅₀: 4.57 and 2.25 μM) is lower than that of angiogenin (**Figure 8**).

In summary, these findings suggest that while RNASE4 exhibits a broad-spectrum antimicrobial activity against Gram-negative bacteria, its inhibitory effect on *Parasutterella* is particularly pronounced compared to other tested bacteria. The differential activities of RNASE A superfamily members against various bacterial species highlight the importance of investigating the specific interactions between individual RNASEs and their target bacteria to better understand their roles in regulating the gut microbiota and host health.

Figure 8 Inhibitory effects of RNASE4 on Gram-negative and Gram-positive bacteria.

8) Have *Rnase4*^{-/-} mice microbial phenotypes beyond the colonic gut microbiota? Also in the small intestine and skin or is it relevant only in the colon?

Response: It is an interesting question. To address your concern, we have indeed analyzed the expression of RNASE4 in different tissues and found that while it is most highly expressed in the colon, there are also notable expression in the small intestine and skin. In Fig. 1 of the manuscript, we have already presented the expression and distribution of RNASE4 in the colon and small intestine.

Additionally, we examined RNASE4 expression in the skin and observed that it is primarily localized in the outer layers of the epidermis and hair follicles (**Figure 9**), suggesting a potential role in regulating the skin microbiota.

Following your suggestion, we further investigated the microbial communities in the small intestine and skin of *Rnase4*^{-/-} mice (**Figure 9**). In the small intestine, we focused on analyzing the bacteria in the mucus layer and found that *Rnase4* knockout does change the composition of microbiota. Similarly, our analysis of the skin microbiota yielded comparable results. Our data suggest that RNASE4 may play its antimicrobial role in various tissues.

However, as the main focus of our present study was on the gut microbiota and its association with colitis, we did not incorporate the aforementioned results into the main manuscript.

Nevertheless, your valuable input has prompted us to explore even broader implications of RNASE4 deficiency on microbial communities beyond the colon. These data will guide our future research to further elucidate the multifaceted roles of RNASE4 in host-microbe interactions across various body sites.

Figure 9 RNASE4 regulates the diversity of intestinal mucosal and skin microbiota.

a Expression levels of RNASE4 in colon, rectum, small intestine and skin based on data from the Human Protein Atlas (www.proteinatlas.org). **b** Localization of RNASE4 in mouse skin tissue. Black arrows indicate RNASE4 positive staining. **c-d** Analysis of microbiota diversity in small intestinal mucosa and skin. Left: bacterial α -diversity within each group; Middle: β -diversity of microbiota composition; Right: dissimilarity analysis within and between groups.

9) Was the RNASE4 knockout mouse also tested in other facility? Can the author rule out that this effect might be due to specific housing (and a yet undefined pathobiont)?

Response: You've raised a valid concern that is commonly encountered in gut microbiota research. To address your concern, we have contacted with the animal center and got to know that our animal facility conducts regular health monitoring every month to ensure there is no contamination by specific pathogens. Secondly, to rule out the potential influence of the housing environment on the mice, we have analyzed the gut microbiota changes in mice housed in two different rooms within the same facility and compared them with mice housed in a separate facility located in Liangzhu Laboratory (**Figure 10**). The results were similar in all the environments. Therefore, we can confidently exclude the housing conditions as a factor affecting the gut microbiota in the *Rnase4* knockout mice.

Figure 10 Analysis of gut microbiota differences in mice housed in different animal facilities.

Reviewer #3 (Defensin/AMP, microbiota, IBD)

General comments:

The authors of the manuscript hypothesized for the first time that RNASE4 regulates the gut microbiota as an antimicrobial protein and showed that RNASE4 expressed in certain gut epithelial cells including goblet cell and Paneth cell have some effects on shaping the gut microbiota as well as a protective role in DSS colitis via *Parasutterella*. The authors also claimed that RNASE4 is an antimicrobial protein that maintains microbial balance and prevents inflammation in the gut by a series of mouse experiments and analyzing samples from IBD patients, further may serve as intervention agent for IBD patients. The concept of the study itself is quite new and interesting in the related field. However, there are a series of concerns especially in the study design, materials, interpretation of results, and discussion. Many of those are critical. Major issues are shown below along with Minor issues.

Specific comments:

- 1) The authors should define antimicrobial peptide (AMP) following current general understandings of those. Despite the authors described that RNASE4 is an antimicrobial protein but not a peptide, there are contamination in use of the term AMP everywhere in the manuscript. In addition, it is inappropriate to refer to the products/secretions of the two cells, goblet cell and Paneth cell, together as AMPs. To date, AMPs in mammalian small intestinal epithelial cells especially mice and humans are secreted almost exclusively from Paneth cells, but not goblet cells and the others. More importantly, goblet cells secrete mucins that harbor the gut microbiota, but not AMPs. All the inadequate terms and sentences should be corrected.

Response: We appreciate your valuable comments and suggestions. You are right, using the term “AMP” interchangeably for both antimicrobial peptides and proteins is inappropriate. To address this issue, we have revised the introduction to focus on antimicrobial proteins secreted by intestinal epithelial cells and have carefully reviewed the entire manuscript to ensure the correct usage of this term.

Regarding the secretion of antimicrobial factors by different cell types in the intestinal epithelium, we highly value your comment on the distinct roles of goblet cells and Paneth cells. While it is true that antimicrobial peptides are primarily secreted by Paneth cells in the small intestine, some

antimicrobial proteins can be secreted by both Paneth cells and goblet cells. For example, RELM β is predominantly produced by goblet cells (*Proc Natl Acad Sci U S A*, 2017;114(42):11027-11033). Similarly, angiogenin (ANG), an anti-microbial protein, is secreted by goblet cells in the large intestine post *Trichuris muris* infection (*PLoS One*, 2012; 7(9): e42248). To address your concern, we have removed the description about the specific cell types that secrete AMPs in the introduction, and re-evaluated our results to objectively described RNASE4 as an antimicrobial protein expressed by intestinal epithelial cells.

We hope that this revised introduction (please refer to line 44-51 of the revised manuscript) provides a more accurate and objective context for our study, while still highlighting the importance of antimicrobial proteins in maintaining gut homeostasis.

2) The authors showed that both goblet and Paneth cells are immunostained with RNASE4 and described that RNASE4 is secreted from both cells to contribute immunity. However, the issue important here is that we don't know all the effects shown in the manuscript came from which cell or even the other cell. Where are the fecal RNASE4 coming from? The authors only showed the immunostainings of RNASE4, then, described that feces and colonic mucosal residues contain RNASE4 in some level. However, the authors failed to provide clear evidence that it is secreted from goblet cells and/or Paneth cells. In addition, if these cells secrete RNASE4, a number of important questions related to the underlying mechanism raises. For example, how the cells or cell secrete RNASE4? What kind of stimuli let them secrete? At this point, the authors failed to provide the underlying mechanism for observed results in the manuscript. Thus, to support their conclusions, carefully designed additional experiments such as cultured intestinal organoids and/or other methods would be needed. Note that so many cell lineages in the intestine express RNASE4.

Response: We greatly appreciate your insightful comment and suggestion regarding the source of RNASE4. Following your instruction, we cultured intestinal organoids, and confirmed that *Rnase4* is expressed in the intestinal epithelium with a notable enrichment in secretory, granule-like structures of goblet and Paneth cells (please see the Fig. 1e of the revised manuscript).

Therefore, currently we have: 1) RNASE4 is expressed in intestinal epithelial cells (predominantly in goblet and Paneth cells), and can be detected in stool samples; 2) *Rnase4* whole-body KO and

intestinal epithelial-specific KO do not change intestinal structure and cell composition, and the morphology of secretory granules or cell polarity of Paneth and goblet cells; 3) The data from *Rnase4*^{ΔIEC} mice indicates the majority of gut RNASE4 is secreted from the intestinal epithelium; 4) *Rnase4*^{ΔIEC} mice have altered gut microbiota and develop severe colitis upon DSS treatment, indicating it is the intestinal epithelium-derived RNASE4 responsible for the observed effects. These results suggest that intestinal RNASE4 accounts for the observed effects and the fecal RNASE4 is derived from intestinal epithelial cells. Unfortunately, we do not have goblet- and Paneth-specific KO mice in hand, and cannot determine RNASE4-originated intestinal cell type(s) at this stage.

We think this work mainly focuses on the regulation of intestinal RNASE4 on gut microbiota and subsequently specific target bacteria, and their contribution to intestinal inflammation. The exact cellular origin of intestinal RNASE4, the stimulus and corresponding regulation mechanisms of its expression and secretion need further study. We hope to get your understanding.

To address your concern, we have provided a more precise description of RNASE4 expression patterns in the revised manuscript.

3) In this study, two genera, *Mucospirillum* and *Parasutterella* were most enriched in *RNASE4*^{-/-}, in addition to decreasing diversity. Then, the increased *Parasutterella* only was highlighted as dysbiosis related with the DSS colitis and IBD. The statements about dysbiosis in both Results and Discussion in the manuscript, especially in relation to experimental colitis and IBD remind us some critical questions. Whether the dysbiosis has links to the patients with UC or CD and how these bacteria contribute to the pathology in the colon and ileum, etc. The authors need to explain such connections if any. Most importantly, much detail and precise evaluation of the gut microbiota is mandatory.

Response: Thank you for raising these critical points. As you may have already noticed, we have done a more detailed and precise evaluation of the gut microbiota following the suggestion from other reviewers as well, including: 1) A detailed description of the changes in alpha and beta diversity indices, along with the appropriate statistical tests and significance levels. 2) A more in-depth analysis of the taxonomic changes at various levels (phylum, class, order, family, and genus), with a focus on the biologically relevant and significantly altered taxa.

Regarding the link between dysbiosis and patients with UC or CD, several studies have shown a significant increase of *Parasutterella* abundance in CD patients (*Clin Exp Gastroenterol*, 2012;5:173, *Inflamm Bowel Dis*. 2023;29(7):1118). *Mucispirillum* has been reported to be increased during inflammation and promote the development of colitis through degrading mucin (*Int J Syst Evol Microbiol*. 2005;55:1199, *ISME J*. 2012;6(11):209). These reports and our work together established the relationship between dysbiosis and experimental colitis/IBD. It is a challenging question regarding how these bacteria contribute to the pathology of intestinal inflammation. Our study provides a mechanistic explanation, i.e., *Parasutterella* suppresses *IDOI* expression and alters tryptophan metabolism in the host cells.

To address your concern, we have incorporated the detailed microbiota analysis in the RESULTS (please refer to line 98-113 of the revised manuscript) and provided a more comprehensive DISCUSSION in the revised manuscript (please refer to line 251-255 of the revised manuscript) to better explain the connection between the observed dysbiosis and IBD pathology.

4) The authors showed killing effects of RNASE4 only against *Parasutterella*, but not other commensal species in the gut. Does RNASE4 have bactericidal activities against gut pathogenic bacteria such as Gram-negative *E. coli* and Gram-positives *S. aureus* as well as other major commensal bacteria such as Lactobacillus, Clostridium, and Bacteroidetes? To understand the gut microbiota in their ecosystem, not just focus on *Parasutterella* but answering these questions is very important, so that further give insights in the underlying mechanism how RNASE4 shapes the composition of gut microbes.

Response: We appreciate you for bringing this crucial point to our attention. Actually, we have indeed investigated the bactericidal activities of RNASE4 against a range of gut bacteria, including both pathogenic and commensal species. As shown in **Figure 8** above, RNASE4 exhibited a broad spectrum of inhibitory activities against different Gram-negative bacteria that are commonly found in the gut. The tested Gram-negative bacteria, including pathogenic species such as *Escherichia coli* (SYY89) and *Salmonella enterica*, as well as commensal species like *Akkermansia muciniphila*, *Brevundimonas diminuta*, *Sphingomonas paucimobilis*, and *Bacteroides vulgatus*, showed varying degrees of sensitivity to RNASE4. The LC₅₀ values ranged from 0.37 μ M (*Parasutterella*) to 4.57 μ M

(*Brevundimonas diminuta*), indicating that RNASE4 effectively inhibits the growth of a wide range of gut Gram-negative bacteria.

In contrast, the tested Gram-positive bacteria, which also included both pathogenic species (*Staphylococcus aureus* ATCC502A and *Listeria monocytogenes*) and commensal species (*Anaerostipes* sp., *Blautia* sp., and *Lactobacillus acidophilus*), were generally less sensitive to RNASE4. The LC₅₀ values were higher than those of the Gram-negative ones, ranging from 22.25 μM to 47.54 μM. These results suggest that RNASE4 has a more limited impact on the growth and survival of gut Gram-positive bacteria.

Therefore, the broad inhibitory activity of RNASE4 against Gram-negative bacteria highlights its potential importance in maintaining a healthy gut microbial ecosystem.

5) The authors of the manuscript showed the involvement of RNASE4 in shaping the gut microbiota but did not measure the levels of the major enteric AMPs that function in the gut lumen reported in previous references. That recalls a question whether abnormalities in α-defensins, cryptdins and HD5, and other AMPs such as Reg3γ and angiogenin are responsible for the pathology shown even partially. Since AMPs including α-defensins, cryptdins, HD5, Reg3γ, angiogenin, etc., secreted from Paneth cells have been known to contribute to the gut innate immunity and regulation of the gut microbiota community, the authors should include these points in Discussion and add levels of those in the *RNASE4*^{-/-} as a new result. To exclude the contribution of these representative AMPs to the observed dysbiosis in *RNASE4*^{-/-}, it is interesting to see levels of such representative AMPs in feces of *RNASE4*^{-/-} mice.

Response: Thank you for raising this important point. We agree that the role of other major enteric antimicrobial peptides and proteins in shaping the gut microbiota should be considered when interpreting our results on RNASE4.

We have indeed taken this into account during our study (**Figure 11**). First, we examined the mRNA levels of several common antimicrobial peptides and proteins in colon epithelial cells, including angiogenin (ANG1), Reg3γ, α-defensins (DEFA1), and cathelicidin (CRAMP). Our results showed that the expression of these antimicrobial peptides and proteins was not significantly affected by the absence of *Rnase4*. Furthermore, to directly assess the levels of representative antimicrobial

peptides and proteins in the gut lumen, we performed immunoblotting analysis (dot blot) on fecal samples from both WT and *Rnase4*^{-/-} mice. The results revealed that the levels of these antimicrobial peptides and proteins in the feces were not substantially different between the WT and *Rnase4*^{-/-} mice. Based on these findings, we conclude that the observed dysbiosis in the gut microbiota of *Rnase4* knockout mice is primarily attributable to the absence of *Rnase4* itself, rather than alterations in the levels of other antimicrobial peptides and proteins.

We have included these additional results in the revised manuscript (please refer to line 93-97) and have also addressed your concern in the DISCUSSION section (please refer to line 226-234 of the revised manuscript).

Figure 11 The expression of key antimicrobial genes in intestines.

a and d, Quantitative mRNA expression of the selected antimicrobial genes in the colons, measured by quantitative PCR (n=6). **b-c and e-f**, Immunoblots and their quantification for key antimicrobial proteins in stool samples (n=3). Data are presented as mean ± SEM. Statistical significance was determined by two-tailed unpaired Student's *t*-test.

6) Related with #2, whether the observed intestinal microbiota changes, dysbiosis, was due to abnormalities in Paneth cells or goblet cells or anything else? Because no phenotypic changes were found in both cells and even entire intestine in *RNASE4*^{-/-} mice, the authors tried to explain all the effects seen by the intervention, owing to observed molecular changes in some intestinal epithelial cells. However, a vital question why the dysbiosis in *RNASE4*^{-/-} both without and with intervention occurred remain unknown or too short to be proved, so that we have to say the underlying mechanisms of the effects in *RNASE4*^{-/-} mice is puzzle.

Response: We appreciate that you raise this challenging question. First of all, as stated in the reply to your comment #2, we concluded that the observed effects come from at least the intestinal epithelial cells, although we could not say clearly whether it is Paneth cells or goblet cells. While there are no overt phenotypic changes in Paneth cells, goblet cells, or the entire intestine in *Rnase4*^{-/-} mice, the lack of spontaneous colitis in these KO mice indicates that the loss of *Rnase4* alone may not be sufficient to trigger intestinal inflammation. In other words, *Rnase4* deletion might be a contributing factor rather than a primary cause of intestinal inflammation. Furthermore, we provided a mechanistic explanation on *Rnase4* deficiency-resulted susceptibility to experimental colitis, that is lack of *Rnase4* in gut lumen caused over-growth of certain Gram-negative bacteria including *Parasutterella*, the latter inhibited *IDO1* expression and thus down-regulated tryptophan metabolism. Meanwhile, we explored the antibacterial mechanism of RNASE4 on this bacterium.

Taken the current results together, we think the underlying mechanism of the effects in *Rnase4*^{-/-} mice is overall clear: RNASE4 kills certain Gram-negative bacteria by forming pores on their membrane; its deficiency results in these bacteria overgrowth and gut dysbiosis; one of the bacteria, *Parasutterella*, downregulates tryptophan metabolism in the intestinal epithelium by suppressing *IDO1* expression, thus exacerbating colitis.

We admit that currently we do not know the regulation mechanism underlying RNASE4 expression and secretion in intestinal epithelial cells, nor the specific secretory cell types, as stated in the reply to #2 comment as well.

7) Although the recombinant RNASE4 and the RNASE4 extracted from mouse intestine were used in this study, no molecular information such as amino acids and structure were provided in

Materials and Methods section. So, it is difficult to evaluate qualities of those RNASE4. Need to add detailed characteristics of the RNASE4 used in the manuscript.

Response: Thank you for bringing this oversight to our attention. Since we have been working on this protein for long time and published a series of articles, we may have inadvertently omitted some essential details in the previous manuscript.

To address this concern, we have now added relevant information in the METHODS section (please refer to line 351-366 of the revised manuscript) and included new data in Supplementary Fig. 17, which showcases the crystal structure of RNASE4, the sequence alignment of human and mouse RNASE4, SDS-PAGE analysis of the purified recombinant protein, and the ribonucleolytic activity of RNASE4 and its variant RNASE4-K40A on total RNA and yeast tRNA substrates.

8) In the manuscript, the authors just stated that no phenotype was found in *RNASE4*^{-/-} mice on the intestinal epithelial cells including goblet and Paneth cells. How far did the authors investigate before reaching that conclusion? It is important to know whether the structure of the intestinal epithelium, especially the morphology of secretory granules and cell polarity are also normal. Since the granule shape itself has been reported to be very important especially for Paneth cells in terms of functions including Paneth cells in CD, the authors should conduct appropriate analyses to confirm this issue.

Response: We appreciate your instruction to improve our work. According to your recommendation, we have performed transmission electron microscopy (TEM) to examine the ultrastructure of goblet cells and Paneth cells in the intestinal epithelia of both WT and *Rnase4*^{-/-} mice (**Figure 12**).

Figure 12 The representative TEM images of goblet and Paneth cells.

The TEM images showed no notable differences in the overall structure of the intestinal epithelium between WT and *Rnase4*^{-/-} mice. Specifically, in goblet cells, the secretory mucus granules were properly located in the apical region of the cells, with no apparent alterations in their size, shape, or electron density. Similarly, in Paneth cells, the secretory granules were situated at the apical region of the cells, with no evident changes in their morphology or distribution. While we acknowledge that there might be subtle molecular changes that are not detectable by TEM, our current observations suggest that *Rnase4* knockout does not significantly impact the overall structure of the intestinal epithelium, including the morphology of secretory granules and cell polarity in goblet cells and Paneth cells.

We have included these additional results in the revised manuscript (please refer to Supplementary Fig.4). We hope that these data, along with our previous observations, support the conclusion that the phenotypes observed in *Rnase4*^{-/-} mice are primarily due to the direct effects of RNASE4 on the gut microbiota rather than secondary effects on intestinal epithelial cell structure and function.

9) It was reported that in angiogenin^{-/-} mice, Lachnospiraceae decreased while α -Proteobacteria increased. It is interesting to know similar changes happened in RNASE4^{-/-} mice.

Response: Thank you very much for your attention on our published Gut paper, and raising an interesting point regarding the impacts of angiogenin and RNASE4 on microbiota. To address this issue, we reanalyzed the bacterial changes in *Rnase4*^{-/-} mice, and found that there were no significant differences in Lachnospiraceae and α -Proteobacteria (**Figure 13a-b**).

Then, we investigated whether there were any pair of bacteria exhibiting a reciprocal relationship in the *Rnase4^{-/-}* mice, as found between Lachnospiraceae and α -Proteobacteria in *angiogenin^{-/-}* mice. Indeed, there was decreased abundance of certain gut bacteria in the fecal samples of *Rnase4^{-/-}* mice, particularly *Akkermansia*. Therefore, we performed experiments to investigate the potential relationship between *Parasutterella* and *Akkermansia* (Figure 13c-f). In our *Parasutterella* transplantation experiments, we found that the abundance of *Akkermansia* was not significantly suppressed in mice receiving *Parasutterella* compared to the control, suggesting that the presence of *Parasutterella* does not directly lead to a decrease in *Akkermansia* *in vivo*. We also conducted *in vitro* experiments in which we cultured *Parasutterella* and *Akkermansia* separately in the presence of each other's culture supernatants. These results showed no apparent antagonistic effects between the two bacteria. These findings are consistent with the current literature on *Parasutterella* (*ISME J*, 2019;13(6):1520), which indicates that this bacterium does not significantly alter the growth of other gut microbes.

While the decrease in *Akkermansia* in *Rnase4^{-/-}* mice is intriguing and warrants further investigation, our current data suggest that this change is unlikely to be directly mediated by the presence or absence of *Parasutterella*. Therefore, we have not included this aspect in our manuscript.

Figure 13 Analysis of the relationship between *Parasutterella* and *Akkermansia* in *Rnase4^{-/-}* mice. a-c Relative abundance of Lachnospiraceae, α -Proteobacteria and *Akkermansia* in WT and *Rnase4^{-/-}* mice. d Relative

abundance of *Akkermansia* in mice transplanted with *Parasutterella* compared to control mice. **e-f** In vitro growth of *Parasutterella* (e) and *Akkermansia* (f) in the presence of each other's culture supernatants. Data are presented as mean \pm SEM. Statistical significance was determined by two-tailed unpaired Student's *t*-test.

10) Page 8 and Fig 4e: It is hard to find positive meanings on that fecal RNASE4 protein expression in IBD patients was 1.4-fold lower than control subjects. What do the authors try to say by that?

Response: Thank you for raising this important point. Although the observed difference in fecal RNASE4 levels between IBD patients and control subjects may not appear to be substantial, our findings from animal models and *in vitro* experiments strongly support the protective role of RNASE4 in colitis and its potential medical applications. It is worth noting that the fecal RNASE4 levels in our study were measured at a single time point in a cross-sectional IBD cohort. To better understand the dynamic changes in RNASE4 expression and its relationship to disease activity, longitudinal studies with repeated measurements of fecal RNASE4 levels over time would be highly informative. Moreover, in future studies, we plan to evaluate RNASE4 levels in combination with other established biomarkers of gut inflammation, such as fecal calprotectin and cytokines, as you mentioned later. We have now integrated these points into the DISCUSSION section of our revised manuscript (please refer to line 265-276).

11) Page 11: Methods in diagnosis of IBD has been largely developing these days. The sentences in Discussion section about IBD diagnosis was not quite right, neglecting recent references such as calprotectin and cytokines.

Response: Thank you for your insightful comment regarding IBD diagnosis. In light of your instruction and the points raised in the previous comment, we have now revised the DISCUSSION section to better reflect the current progress and the potential contribution of RNASE4 in IBD diagnosis (please refer to line 265-276 of the revised manuscript).

12) The authors administered RNASE4 orally for the treatment of DSS colitis in the manuscript and Extended Data Fig.12 but did not show that administered RNASE4 was gone through the intestinal lumen and elicit the effects. Did the authors check RNASE4 concentration in the intestinal lumen

or the feces during RNASE4 administration? It should be added because this could be critical data to support their conclusion.

Response: Thank you for your valuable comment and suggestion. Following your instruction, we employed a semi-quantitative immunoblotting method to detect RNASE4 concentration in mouse samples due to the lack of commercially available ELISA kits. As shown in **Figure 14**, the RNASE4 concentration in the small intestinal lumen of WT mice was found to be approximately 0.51 μM (or about 0.38 $\mu\text{g/g}$ total protein). Following oral administration of RNASE4, the level of RNASE4 in the intestinal lumen of *Rnase4^{-/-}* mice was around 0.62 μM (or about 0.46 $\mu\text{g/g}$ total protein). The RNASE4 concentration in the feces of WT mice was approximately 0.11 μM (or about 0.08 $\mu\text{g/g}$ total protein), while the level of RNASE4 in the feces of *Rnase4^{-/-}* mice after oral administration was around 0.44 μM (or about 0.33 $\mu\text{g/g}$ total protein). These results provide evidence that orally administered RNASE4 can indeed reach the intestinal lumen and be detected in the feces.

It is worth pointing out these concentrations are close to the LC_{50} value (0.37 μM) for the *in vitro* inhibition of *Parasutterella* growth, suggesting that the administered RNASE4 can effectively suppress the growth of this bacterial genus.

a

b
Figure 14. RNASE4 level in the small intestinal lumen and feces of mice during RNASE4 administration.

a-b Immunoblotting analysis of RNASE4 protein level in the small intestinal lumen or feces from WT mice and *Rnase4*^{-/-} mice treated with or without RNASE4. The standard curve on the right shows the linear relationship between RNASE4 concentration and gray value. Table showing the gray values, RNASE4 concentrations (ng/μl), RNASE4 concentrations in the lumen or feces (μM, based on RNASE4 molecular weight of 14.7 kDa), and RNASE4 concentrations in the lumen (μg/g total protein) for each sample.

13) The authors showed mRNA expression of several potentially functional molecules including cytokines. How about the expression of protein levels of those molecules?

Response: Thank you for your valuable suggestion regarding the detection of the cytokines. Following your requirement, we have analyzed the protein concentrations of the relevant inflammatory factors in the DSS-induced and TNBS-induced colitis models by ELISA, and found that they were consistent with their corresponding mRNA levels. Specifically, the protein levels of CCL3, CXCL1, CXCL2, IL-6, IL-1β, and TNFα were upregulated in *Rnase4* knockout colitis mice (Fig.3h and Supplementary Fig.6 of the revised manuscript, **Figure 15**).

Figure 15. Quantitative protein level of indicated cytokines in the colons of WT and *Rnase4*^{-/-} mice after DSS (a) or TNBS (b) treatment

14) It would be interesting to see what happened in the systemic immunity in *RNASE4*^{-/-} mice? Is the dysbiosis and the susceptibility to DSS colitis occurred in *RNASE4*^{-/-} mice simply because leaky gut? Or are there any further effects in innate and adaptive immunity?

Response: We appreciate your valuable suggestion to explore the systemic immunity in *Rnase4*^{-/-} mice. To address your query, we have conducted experiments to assess the levels of innate immune cells and T cells in the spleen and mesenteric lymph nodes (MLNs) of *Rnase4*^{-/-} mice. As shown in **Figure 16**, the absence of *Rnase4* did not significantly alter the proportions of these immune cell populations in both sites. These data suggest that the dysbiosis and increased susceptibility to DSS colitis in *Rnase4*^{-/-} mice may not be primarily due to alterations in systemic immune cell composition. Furthermore, our results from intestinal epithelial cell-specific *Rnase4* knockout mice revealed that the loss of *Rnase4* in epithelial cells is sufficient to induce gut microbiota imbalance and increase colitis susceptibility. These findings highlight the critical role of epithelial cell-derived RNASE4 in regulating the gut microbiome and protecting intestinal integrity.

Figure 16. *RNASE4* deficiency does not alter systemic immune cell composition but affects local immune responses in the gut.

a Representative image of spleen and mesenteric lymph nodes (MLNs) from wild-type (WT) and *RNASE4*^{-/-} mice. **b** Spleen weight and MLN weight in WT and *RNASE4*^{-/-} mice (n = 4 per group). **c** Flow cytometric analysis of immune cell populations in the spleen and MLNs of WT and *RNASE4*^{-/-} mice. The plots show the gating strategies and percentages of macrophages (CD11b⁺ F4/80⁺), dendritic cells (DCs; CD11b⁺ CD11c⁺), and myeloid-derived suppressor cells (MDSCs; CD11b⁺ Gr-1⁺). **d** Quantification of macrophages, DCs, and MDSCs in the spleen and MLNs of WT and *RNASE4*^{-/-} mice (n = 4 per group). **e** Flow cytometric analysis of T cell populations in the spleen and MLNs of WT and *RNASE4*^{-/-} mice. The plots show the gating strategies and percentages of CD4⁺ and CD8⁺ T cells. **f** Quantification of CD4⁺ and CD8⁺ T cells in the spleen and MLNs of WT and *RNASE4*^{-/-} mice (n = 4 per group). Data are presented as mean ± SEM.

15) Discussion and References: The authors missed many important previous studies to discuss and cite. Discussion section is insufficient and need to reconstruct, partially due to lack of previous important research results and logical configuration. References shown in the manuscript are inadequate and many important studies are missing, so it should be thoroughly checked and revised.

Response: We appreciate your critical comments on the Discussion section. To address this limitation, we have conducted a comprehensive literature review and extensively revised the Discussion section. The restructured Discussion section now addresses the following key points:

- 1) A summary of our findings and potential significance;
- 2) Based on overview of intestinal antimicrobial proteins and their general antimicrobial mechanism, discuss the action characteristics of RNASE4 in intestine and point out the undissolved challenges in our study.
- 3) An in-depth analysis of the identified bacterium *Parasutterella*, focusing on its role within the gut microbiome and its contribution to intestinal inflammation.
- 4) An introduction of potential medical applications of RNASE4, particularly in the diagnosis and management of IBD.

We think that the reconstructed discussion has now updated research progress in this field, cited the necessary literature, and analyzed our study limitations, and hope you are satisfying with it.

16) Most Figures of histology with HE staining and immunohistochemistry need to replace to clear images or at least add photos with high magnification views, because the important parts affecting credibility of the results are not clear.

Response: Thank you for your friendly reminder. We have carefully reviewed the images and replaced the previous ones with clear photographs of higher-quality. Additionally, we have provided high-magnification views of the critical regions for better understanding of our results. We hope that the revised figures adequately address your concern and meet the standard required for publication.

REVIEWER COMMENTS

Reviewer #1 (Remarks to the Author):

I thank the authors for their careful and detailed responses to my original comments and I have no further concerns

Reviewer #2 (Remarks to the Author):

The authors present a substantially revised manuscript.

Reviewer #3 (Remarks to the Author):

The authors of the manuscript responded to most of reviewers' concerns adding a lot of substantial experiments, and the revised manuscript largely improved. However, the reviewer suggestions and comments partially were not adequately addressed, so that a few but clear concerns yet remain in the revised manuscript.

About cells in the gut producing RNASE4, because "the intestinal epithelial cells" is too unclear, "the intestinal epithelial cells including Paneth cells and goblet cells" is rather recommended when the authors' findings in this study were maximally incorporated.

Since this study is focusing on an antimicrobial protein RNASE4 contribution to gut mucosal immunity and IBD, it is odd that existing related important AMP studies are still neither sufficiently referred nor discussed in the revised manuscript despite of previous reviewer suggestions. For instance, there is a consensus that α -defensins are the major specific constituents of Paneth cell granules. There are many reports showing that α -defensins, mouse cryptidins and human HD5, play a vital role in the gut mucosal immunity as well as regulation of the gut microbiota composition. Furthermore, abnormalities in α -defensins and some other AMPs have been reported to relate with diseases or disease models including obesity, Crohn's disease, liver disease, aging, etc., via disruption of gut mucosal immunity followed by dysbiosis. Even though the authors used and analyzed RNASE4 in mouse intestine as an antimicrobial protein, which is secreted presumably by Paneth cells and goblet cells in this study, the major AMPs including α -defensins need to be addressed along with RNASE4 or at least discuss detail. Again, that was already pointed out, but not correctly addressed by the authors.

From the limited TEM images, Paneth cell granules in RNASE4 knockout mice seem to have more vacuoles than those in WT mice, so it may be a good idea to analyze key molecules related to ER stress or autophagy in detail in future RNASE4 studies for IBD mechanisms. At this point, this reviewer agrees to assume that there are no significant changes in both cell lineages in terms of morphology. However, because it has been reported, and there is a consensus that abnormalities in granules of Paneth cells in patients with Crohn's disease and the mouse models reflect or due to abnormalities in ER and autophagy, it is recommended to discuss these points in the Discussion section.

Point-by-point Response to Reviewers' Comments

Reviewer #1 (Remarks to the Author):

I thank the authors for their careful and detailed responses to my original comments and I have no further concerns.

Response: Thank you for your positive feedback and support.

Reviewer #2 (Remarks to the Author):

The authors present a substantially revised manuscript.

Response: We sincerely appreciate your recognition of our efforts to improve the manuscript.

Reviewer #3 (Remarks to the Author):

The authors of the manuscript responded to most of reviewers' concerns adding a lot of substantial experiments, and the revised manuscript largely improved. However, the reviewer suggestions and comments partially were not adequately addressed, so that a few but clear concerns yet remain in the revised manuscript.

1) About cells in the gut producing RNASE4, because "the intestinal epithelial cells" is too unclear, "the intestinal epithelial cells including Paneth cells and goblet cells" is rather recommended when the authors' findings in this study were maximally incorporated.

Response: We fully agree with your suggestion, as it better reflects our results. We have modified the corresponding description accordingly in the Abstract section (please refer to line 34 of the revised manuscript).

2) Since this study is focusing on an antimicrobial protein RNASE4 contribution to gut mucosal immunity and IBD, it is odd that existing related important AMP studies are still neither sufficiently referred nor discussed in the revised manuscript despite of previous reviewer suggestions. For instance, there is a consensus that α -defensins are the major specific constituents of Paneth cell granules. There are many reports showing that α -defensins, mouse cryptidins and human HD5, play

a vital role in the gut mucosal immunity as well as regulation of the gut microbiota composition. Furthermore, abnormalities in α -defensins and some other AMPs have been reported to relate with diseases or disease models including obesity, Crohn's disease, liver disease, aging, etc., via disruption of gut mucosal immunity followed by dysbiosis. Even though the authors used and analyzed RNASE4 in mouse intestine as an antimicrobial protein, which is secreted presumably by Paneth cells and goblet cells in this study, the major AMPs including α -defensins need to be addressed along with RNASE4 or at least discuss detail. Again, that was already pointed out, but not correctly addressed by the authors.

Response: We are sorry that our previous revision did not correctly address your comments, and highly appreciate your effort to better improve our manuscript. Actually, we self-righteously thought that you asked us to focus on antimicrobial protein only, thus forgetting to sufficiently refer the existing related important AMP studies. Following your instruction, we have now incorporated an overview about the major intestinal AMPs including α -defensins and give a prospect regarding the role of RNASE4 and their potential relationship in this secondly revised manuscript (please refer to line 226-236 of the revised manuscript). We hope that the discussion about AMPs is comprehensive, and you are satisfying with the revision.

3) From the limited TEM images, Paneth cell granules in RNASE4 knockout mice seem to have more vacuoles than those in WT mice, so it may be a good idea to analyze key molecules related to ER stress or autophagy in detail in future RNASE4 studies for IBD mechanisms. At this point, this reviewer agrees to assume that there are no significant changes in both cell lineages in terms of morphology. However, because it has been reported, and there is a consensus that abnormalities in granules of Paneth cells in patients with Crohn's disease and the mouse models reflect or due to abnormalities in ER and autophagy, it is recommended to discuss these points in the Discussion section.

Response: Thank you for your insightful comment and suggestion. Following your recommendation, we have highlighted the existed phenomenon of TEM images in the Discussion section, and have further discussed that RNASE4 may influence the formation and maturation of secretory vesicles in Paneth cells and the potential causes (please refer to line 236-245 of the newly revised manuscript). We have also referenced the established link between abnormalities in Paneth cell granules and IBD

pathogenesis (*Gastroenterology* 125, 47-57 (2003); *Front Immunol* 11, 646 (2020)), as well as the reported disturbances in ER function and autophagy pathways in these disorders (*Nature* 456, 259-263 (2008); *Cell* 134, 743-756 (2008)). Given this consensus, we have emphasized the need for future studies to investigate the potential contribution of ER stress and autophagy along with RNASE4 in IBD mechanisms. We hope that the addition of this discussion in the secondly revised manuscript could address your concern.

REVIEWERS' COMMENTS

Reviewer #3 (Remarks to the Author):

I have no further suggestions and comments.